# Mask Image Watermarking

**Runyi Hu**[1], **Jie Zhang**[2]*, **Shiqian Zhao**[1], **Nils Lukas**[3],
**Jiwei Li**[4], **Qing Guo**[2], **Han Qiu**[5], **Tianwei Zhang**[1]
[1]Nanyang Technological University    [2]CFAR and IHPC, A*STAR, Singapore
[3]MBZUAI    [4]Zhejiang University    [5]Tsinghua University
{runyi.hu, tianwei.zhang}@ntu.edu.sg
zhangj6@a-star.edu.sg
https://github.com/hurunyi/MaskWM

## Abstract

We present MaskWM, a simple, efficient, and flexible framework for image watermarking. MaskWM has two variants: (1) MaskWM-D, which supports global watermark embedding, watermark localization, and local watermark extraction for applications such as tamper detection; (2) MaskWM-ED, which focuses on local watermark embedding and extraction, offering enhanced robustness in small regions to support fine-grined image protection. MaskWM-D builds on the classical *encoder-distortion layer-decoder* training paradigm. In MaskWM-D, we introduce a simple masking mechanism during the decoding stage that enables both global and local watermark extraction. During training, the decoder is guided by various types of masks applied to watermarked images before extraction, helping it learn to localize watermarks and extract them from the corresponding local areas. MaskWM-ED extends this design by incorporating the mask into the encoding stage as well, guiding the encoder to embed the watermark in designated local regions, which improves robustness under regional attacks. Extensive experiments show that MaskWM achieves state-of-the-art performance in global and local watermark extraction, watermark localization, and multi-watermark embedding. It outperforms all existing baselines, including the recent leading model WAM for local watermarking, while preserving high visual quality of the watermarked images. In addition, MaskWM is highly efficient and adaptable. It requires only 20 hours of training on a single A6000 GPU, achieving 15× computational efficiency compared to WAM. By simply adjusting the distortion layer, MaskWM can be quickly fine-tuned to meet varying robustness requirements.

## 1 Introduction

Image watermarking [15] is a crucial technique for embedding imperceptible information into images, serving purposes such as copyright protection, content authentication, and provenance tracking. With the proliferation of AI-generated content (AIGC) [17, 19], the boundary between real and synthetic images has become increasingly blurred, making it especially important to develop robust watermarking schemes for content verification and traceability.

Traditional deep image watermarking methods [32, 22, 10, 4, 26] typically perform global watermark embedding and extraction, treating the entire image as a uniform entity. However, these global approaches suffer from several critical limitations. 1. *Lack of local watermark extraction*: When an image undergoes heavy tampering, such as inpainting [25, 28], the watermark may survive only in a small, local region that remains untouched by manipulation. In such cases, global methods often fail

---

*The corresponding author

to extract the watermark effectively. 2. *Inability to localize the watermark*: Even if a watermark is successfully extracted from the image, global methods cannot localize which region actually contains the watermark, making it difficult for fine-grained forensic analysis and fair judgment. 3. *Lack of local watermark embedding*: In scenarios where only specific regions of an image are valuable and need protection, or when different parts of the image originate from different sources and require distinct watermarking, global embedding is inherently incapable of providing the flexibility and granularity.

We argue that the training paradigm of traditional global watermarking methods, which treats the entire watermarked image as a whole for both encoding and decoding, prevents the encoder and decoder from developing spatial awareness. Specifically, the decoder cannot identify which regions of the image contain watermark signals for effective extraction, while the encoder lacks the ability to adaptively embed the watermark into specific spatial regions.

Based on the above analysis, we propose MaskWM, a simple, efficient, and flexible image watermarking framework. MaskWM introduces a masking mechanism during training to guide the model learn spatially aware embedding and extraction of local watermark signals. Depending on the stage which the mask is introduced, MaskWM offers two variants: **MaskWM-D** introduces the mask only during the decoding phase, enabling global watermark embedding while supporting local extraction. Specifically, by applying masks to retain only the selected regions of the watermarked images during extraction, the decoder is guided to identify which regions contain watermark signals and to focus on them for effective local extraction. **MaskWM-ED** introduces the mask during the encoding and decoding phases, allowing the embedding and extraction of the local watermark. In this setting, the encoder is trained to embed not only the watermark bits but also the spatial mask into the image. This allows the encoder to leverage the mask to adaptively allocate watermark strength to designated regions, while keeping the rest of the image nearly unaffected.

Extensive experiments demonstrate that MaskWM significantly outperforms existing baselines in both global and local watermark extraction, watermark localization, and local watermark embedding, while preserving image quality. Specifically, for local watermark extraction, MaskWM achieves a nearly 100% extraction accuracy even when only 5% of the image carries watermark signals. In terms of watermark localization, MaskWM demonstrates high precision in identifying watermark regions. Furthermore, although not explicitly trained for multi-watermark embedding, MaskWM maintains strong extraction and localization performance even when embedding up to 5 distinct watermarks in a single image. More importantly, MaskWM exhibits strong robustness across a wide range of distortions, including geometric distortions that typically break many existing watermarking methods. In addition to its effectiveness, MaskWM is highly efficient. Training the encoder-decoder model requires only approximately 20 hours on a single A6000 GPU, which is 15× less compute than the recent state-of-the-art local watermarking model WAM [20]. MaskWM also scales effortlessly to different bit lengths (e.g., 32, 64, and 128), whereas WAM is inherently limited to 32-bit embedding and does not scale beyond that. Furthermore, MaskWM supports fast fine-tuning for different use cases. For example, MaskWM can reach a near 100% extraction accuracy against VAE-based adaptive attacks after just 20k training steps. These advantages make MaskWM a practical, efficient, and scalable solution for real-world applications.

## 2 Background

### 2.1 Image Watermarking

Image watermarking techniques can generally be categorized into two types: *global watermarking* and *local watermarking* methods. **Global watermarking methods aim to extract watermark information from the entire image.** Most traditional deep learning-based approaches fall into this category. These methods focus on achieving robustness against various types of distortions, ensuring that the embedded watermark can still be reliably recovered even when the image undergoes degradation. For example, MBRS [10] specifically targets robustness against JPEG compression. Methods like StegaStamp [22] and PIMoG [5] are designed to handle real-world physical distortions such as screen-shooting and print-shooting. More recent approaches like ZoDiac [27] and SuperMark [6] tackle adaptive attacks, while Robust-Wide [8] and VINE [14] focus on robustness against instruction-driven image editing.

**In contrast, local watermarking methods focus on extracting watermark information from a specific region of the image.** Recent methods, such as WAM [20] and our proposed MaskWM, belong to this category. WAM treats watermark extraction as a segmentation task [12], predicting watermark bits at the pixel level and then averaging these per-pixel predictions to obtain the final result. While this fine-grained approach enables local watermark extraction, it also presents several challenges. 1. *Limited extraction from small regions*: When the watermarked area is very small, only a few pixels contribute to extraction, making naive averaging unreliable, especially under distortions. 2. *Lack of scalability with longer messages*: WAM struggles to scale beyond 32-bit messages, as training becomes increasingly difficult with longer bit lengths. 3. *High computational cost*: Training WAM is resource-intensive, requiring eight V100 GPUs for nearly a week, which limits its practicality. 4. *Lack of native local embedding*: WAM embeds watermarks globally and then crops for local focus, introducing embedding losses that reduce extraction robustness.

## 2.2 Watermark Localization

Watermark localization [30, 7, 20] refers to the ability to determine which regions of a watermarked image still contain watermark information after modifications. This capability enables the identification of unaltered content, serving as an active detection mechanism for tamper localization. Currently, image watermarking techniques primarily adopt two paradigms for watermark localization. **The first paradigm embeds a one-dimensional copyright watermark alongside a two-dimensional localization watermark in the original image.** During extraction, localization is based on the fragility of the localization watermark, which cannot be fully recovered from a modified image. Key methods in this category include EditGuard [30] and OmniGuard [31]. EditGuard embeds a solid-color template within the host image and attempts to recover it from a modified version. The difference between the recovered and original templates is calculated at each pixel, and a threshold-based decision identifies watermark-preserved regions. OmniGuard improves upon EditGuard by embedding a natural image as the template, enhancing fidelity. It also introduces a Degradation-aware Tamper Extractor, improving robustness in detecting tampered regions under distortion. This paradigm requires parallel extraction of both copyright and localization watermarks, which may affect image quality. Moreover, both watermarks need independent robustness, and the presence of the template watermark does not guarantee the presence of the copyright watermark.

**The second paradigm, in contrast, embeds only a one-dimensional copyright bitstream and directly determines the presence or absence of watermark information at each pixel to achieve localization.** Methods such as WAM [20] and our proposed MaskWM fall under this category. WAM employs a decoder that simultaneously performs pixel-wise watermark presence detection and copyright bit extraction. In contrast, our MaskWM incorporates a dedicated localization module within the decoder, focusing solely on watermark presence detection at each pixel. This approach is more lightweight and easier to optimize. Compared to the first watermark localization paradigm, this method ensures that the local watermarked regions strictly correspond to the areas containing copyright watermark information, enhancing interpretability. Additionally, it guarantees both the robustness of copyright watermark extraction and the robustness of localization.

## 3 Methodology

### 3.1 Design Principles

In general, we have three main objectives: **local watermark extraction**, **watermark localization**, and **local watermark embedding**. Among these, our primary goal is local watermark extraction, which aims to recover the embedded message from images where only a small, spatially local region contains the watermark signal. In practice, we find that achieving high performance on this task naturally necessitates solving the other two problems as well. We now identify three key reasons why traditional watermark models fail under this setting.

*First*, since the decoder is trained exclusively on globally watermarked images and has never encountered cases with only locally embedded watermarks, it fails to perform zero-shot extraction on such inputs. *Second*, the non-watermarked portions of an image interfere with the decoder's extraction process, especially when the watermark occupies a small area, leading to extraction failure. *Third*, because the decoder is optimized for global watermark extraction, the encoder tends to dilute the

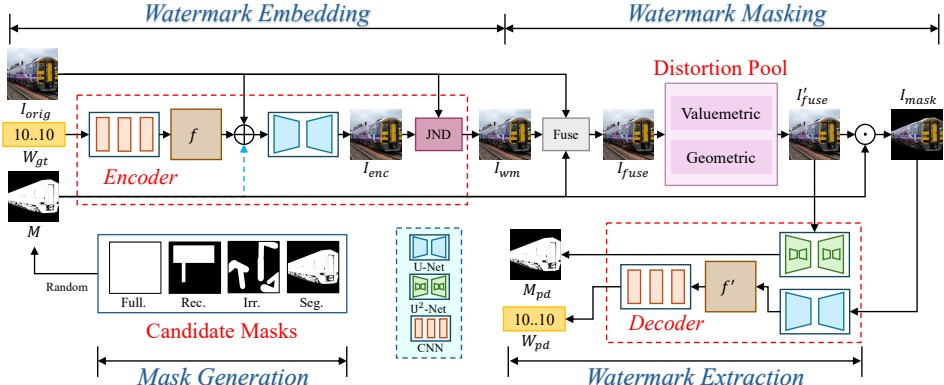

Figure 1: The overall end-to-end training pipeline of MaskWM. (1) In the **Mask Generation** stage, we generate candidate masks from four predefined types and randomly select one mask $M$ for the subsequent stages. (2) In the **Watermark Embedding** stage, the encoder $\mathcal{E}$ embeds the watermark bits $W_{gt}$ into the original image $I_{orig}$, optionally using the mask $M$ (for MaskWM-ED), to produce the watermarked image $I_{wm}$. (3) In the **Watermark Masking** stage, the mask $M$ is used to fuse $I_{orig}$ and $I_{wm}$ (see Eq. 2), resulting in the fused image $I_{fuse}$, which is then subjected to a randomly selected distortion from a predefined distortion pool, yielding the distorted image $I'_{fuse}$. The masked region is then cropped using $M$ to obtain $I_{mask}$. (4) In the **Watermark Extraction** stage, the decoder $\mathcal{D}$ extracts the predicted mask $M_{pd}$ from $I'_{fuse}$ and the predicted watermark bits $W_{pd}$ from $I_{mask}$.

watermark's intensity over the entire image. This results in local regions having either insufficient or fragmented watermark strength, thereby exacerbating the extraction challenge.

To address these challenges, we propose MaskWM-D, which introduces a basic mask mechanism during the *decoding stage* to guide the decoder in identifying and focusing on watermark-containing regions. To solve the first issue, we retain the watermark only in the masked regions and set other regions' pixels to zero, training the decoder to extract watermarks from partially watermarked images. For the second issue, we replace the non-masked regions with the original clean image and add a watermark localization module in the decoder to differentiate between watermarked and non-watermarked areas, reducing interference from irrelevant content. These two strategies directly address the first two issues by enabling the decoder to extract watermarks from local regions, while also indirectly mitigating the third issue by guiding the encoder to evenly distribute the watermark under the end-to-end training, thus facilitating the decoder's extraction process.

To further enhance the encoder's ability to address the third challenge, we propose MaskWM-ED, which incorporates the mask during the *encoding stage* to explicitly guide watermark placement. In MaskWM-ED, the mask is embedded into the image along with the watermark bits during training. This enables the encoder to learn to actively concentrate the watermark within the selected regions based on the embedded mask, thereby further improving the robustness of local watermarking.

### 3.2 Training

The overall end-to-end training pipeline is shown in Figure 1. It consists of four stages: (1) Mask Generation, (2) Watermark Embedding, (3) Watermark Masking, and (4) Watermark Extraction. In the following, we provide a detailed description of each stage.

**Mask Generation.** The mask generation process constructs a pool of candidate masks with diverse types, from which one mask $M$ is randomly selected for each image during training. We follow a similar mask generation strategy as LaMa [21] for image inpainting, utilizing four distinct types of masks: Full Mask, Rectangle Mask, Irregular Mask, and Segment Mask. These masks can enhance the model's ability to handle watermark embedding and extraction under diverse conditions from different perspectives: *Full Mask* enables the model to embed and extract watermarks across the entire image, serving as a fundamental capability. *Rectangle Mask* focuses on regularly shaped local regions, encouraging the model to operate within confined areas of varying sizes. *Irregular Mask* introduces complex, arbitrarily shaped regions to improve robustness in non-uniform contexts. Lastly,

*Segment Mask* targets semantically meaningful areas by leveraging object masks from the MS-COCO dataset [13], helping the model generalize to real-world scenarios.

**Watermark Embedding.** We describe the process of embedding both the watermark bits $W_{gt}$ and the optional mask $M$ into the original image $I_{orig}$. We first randomly sample binary watermark bits $W_{gt} \in 0, 1$ of length $l$, which are transformed into a feature map $f \in \mathbb{R}^{C_f \times H \times W}$ using a lightweight CNN, where $C_f$ is the number of channels in $f$, and $H$ and $W$ denote the height and width of $I_{orig}$, respectively. This CNN consists of a linear layer that maps $W_{gt}$ to a tensor of shape $(1, l, l)$, followed by bilinear interpolation to $(1, H, W)$ and several Conv-Norm-ReLU (CNR) blocks that produce the final feature map $f$.

To embed the watermark, we concatenate the original image $I_{orig}$ with the watermark feature $f$ along the channel dimension. For the MaskWM-D, this results in a tensor of shape $(3 + C_f, H, W)$, promoting global watermark embedding. For the MaskWM-ED, the optional mask $M$ is further concatenated, yielding a tensor of shape $(3 + C_f + 1, H, W)$. The mask guides the model to focus on the selected regions during training, enabling local embedding within those specified areas.

The concatenated tensor is then passed through a U-Net [12] to generate an intermediate encoded image $I_{enc}$. To obtain the final watermarked image $I_{wm}$, we apply a Just-Noticeable-Difference (JND) module [23], which modulates the embedding signal based on human visual sensitivity to enhance the perceptual quality:

$$I_{wm} = I_{orig} + \mu \times \mathrm{JND}(I_{orig}) \times (I_{enc} - I_{orig}), \tag{1}$$

where $\mu$ is the JND scaling factor to control the watermark strength. We explore several strategies to improve the visual quality of the watermarked image and find that JND modulation consistently delivers the best performance (see Appendix D.1 for details).

**Watermark Masking.** We describe how the mask $M$ is used to process the watermarked image $I_{wm}$ for subsequent mask prediction and watermark extraction by the decoder. First, we generate a fused image $I_{fuse}$ by combining $I_{wm}$ and the original image $I_{orig}$, where the unmasked regions are replaced with the corresponding pixels from $I_{orig}$:

$$I_{fuse} = I_{wm} \odot M + I_{orig} \odot (1 - M). \tag{2}$$

Next, a randomly selected distortion from a predefined distortion pool is applied to $I_{fuse}$, producing an augmented image $I'_{fuse}$. This step follows a common practice in traditional watermarking methods to improve robustness against various transformations. Finally, we use the mask $M$ once more to isolate the watermarked regions of $I'_{fuse}$, setting all other pixels to zero to obtain the input $I_{mask}$ for watermark extraction:

$$I_{mask} = I'_{fuse} \odot M. \tag{3}$$

**Watermark Extraction.** We describe how the decoder $\mathcal{D}$ extracts the predicted mask from $I'_{fuse}$ and recovers the watermark bits from $I_{mask}$. To achieve these two objectives, $\mathcal{D}$ consists of two dedicated modules: a U$^2$-Net [16] for mask prediction, and a U-Net [18] followed by a CNN for watermark extraction. Specifically, the U$^2$-Net takes $I'_{fuse}$ as input and predicts a mask $M_{pd}$. Meanwhile, the U-Net processes $I_{mask}$ to produce an intermediate feature $f'$ with shape $(C_f, H, W)$, which is then passed through a CNN to obtain the predicted watermark bits $W_{pd}$. Unlike the CNN used in the encoder $\mathcal{E}$, the CNN in the decoder $\mathcal{D}$ first applies several Conv–Norm–ReLU layers to further extract features from $f'$. The resulting features are then interpolated to a fixed shape of $(1, l, l)$ and subsequently transformed by linear layers into a bit sequence of length $l$, yielding the final watermark prediction $W_{pd}$.

**Training Objectives.** For all loss functions, we use Mean Squared Error (MSE). Specifically, the encoder loss is defined as:

$$\mathcal{L}_{\mathrm{enc}} = \mathcal{L}_{\mathrm{MSE}}(I_{wm}, I_{orig}). \tag{4}$$

Note that we impose constraints only in the pixel space, as we find that this setup, combined with JND modulation, already achieves high visual quality. Introducing constraints in the feature space or using GAN-based losses would negatively impact the overall performance, as discussed in Appendix D.1. The decoder loss is formulated as:

$$\mathcal{L}_{\mathrm{dec}} = \mathcal{L}_{\mathrm{MSE}}(W_{pd}, W_{gt}) + \alpha \mathcal{L}_{\mathrm{MSE}}(M_{pd}, M), \tag{5}$$

where $\alpha$ is a factor controlling the weight of the mask loss. The overall objective function is:

$$\mathcal{L}_{\text{total}} = \beta_{\text{enc}} \cdot \mathcal{L}_{\text{enc}} + \beta_{\text{dec}} \cdot \mathcal{L}_{\text{dec}}, \tag{6}$$

where $\beta_{\text{enc}}$ and $\beta_{\text{dec}}$ are the weights for the encoder and decoder losses, respectively. Compared to conventional watermarking methods, our approach introduces only a mask loss at the decoder stage. As a result, it retains a simple yet effective objective, making it easy to extend.

### 3.3 Inference

The inference process consists of three main stages: watermark embedding, localization, and extraction. Embedding is performed by the encoder. In MaskWM-D, the encoder embeds the watermark bits across the entire image. In MaskWM-ED, users provide both the watermark bits and a mask, allowing the encoder to embed the watermark into specific regions. Lo-

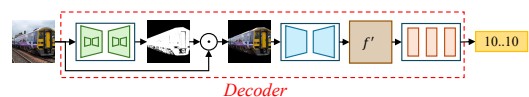
*Decoder*

Figure 2: Watermark localization and extraction process in our decoder during inference.

calization and extraction are performed by the decoder. As shown in Figure 2, unlike conventional global extraction methods, our approach first uses the decoder's localization module to identify watermark-containing regions. The rest of the image is masked out by setting non-watermarked areas to zero, reducing noise during extraction. The watermark is then recovered from the retained regions. As discussed in Sec. 4.5, this strategy improves extraction robustness, especially when only small regions remain watermarked.

### 3.4 Usage Scenarios

This section outlines the usage scenarios of MaskWM-D and MaskWM-ED, which are tailored to different protection requirements. **MaskWM-D** is designed for full-image protection. It enables reliable watermark extraction even when parts of the image are tampered with, making it suitable for scenarios that demand global content integrity and tamper detection, such as copyright enforcement or image provenance verification. **MaskWM-ED**, on the other hand, focuses on protecting specific regions of interest, such as faces, logos, or sensitive content. It allows targeted verification or tracing if these regions are reused or misappropriated, without introducing watermarks across the entire image. This strategy not only supports privacy-aware or content-specific protection, but also tends to provide better robustness within the marked regions.

## 4 Experiments

### 4.1 Implementation Details

**Training.**   For all experiments, we train MaskWM on 83k images from the MS-COCO 2014 training set [13] and the training details are provided in Appendix C.1.1.

**Evaluation.**   To ensure fair comparison, we fix the image resolution to $512 \times 512$. For baseline methods that do not support this resolution, we follow the resolution scaling strategy from TrustMark [2] to interpolate the watermark strength (see Appendix C.2.1), which has been shown to preserve watermarking performance. To ensure comparable visual fidelity across variants, we set the JND scaling factor $\mu$ to 1.3 for MaskWM-D and 1.75 for MaskWM-ED. For robustness evaluation, we separately assess *valuemetric* and *geometric* distortions. For valuemetric robustness, we randomly sample from a set of ten common distortions, including JPEG Compression, Gaussian Filter, Gaussian Noise, Median Filter, Salt&Pepper Noise, Resize, Brightness, Contrast, Hue, and Saturation. For geometric robustness, we randomly sample from three typical transformations: Rotation, Perspective, and Horizontal Flip. These distortions collectively cover the vast majority of real-world transformations that watermarked images are likely to encounter in practical scenarios. Detailed parameter settings for each distortion are provided in Appendix C.2.2. We also evaluate robustness under each specific distortion individually and the detailed results are presented in Appendix D.2.

## 4.2 Global and Local Watermarking Comparison

**Settings.** We compare MaskWM with seven recent open-source watermarking methods, including both global (e.g., StegaStamp [22], SepMark [24], TrustMark [2], EditGuard [30], Robust-Wide [8], VINE [14]) and local (WAM [20]) approaches. We use the clean (EditGuard-C) and degraded (EditGuard-D) variants of EditGuard, and the robust version (VINE-R) of VINE. For global watermarking, we evaluate on 1k images from the MS-COCO 2014 validation set using PSNR, SSIM, and Bit Accuracy, where PSNR and SSIM measure the visual quality of watermarked images, and Bit Accuracy evaluates watermark extraction performance. For local watermarking, we use all 41k validation images and evaluate Bit Accuracy under different watermarked area ratios. By default, MaskWM embeds 32 bits for fair comparison with WAM, though MaskWM supports flexible bit lengths (see Sec. 4.5). See Appendix C.2.3 for more implementation details and evaluation settings.

**Global Watermarking Results.** The global watermarking results for all methods are summarized in Table 1. First, both MaskWM-D and MaskWM-ED achieve high visual fidelity, with PSNR scores above 39.5 and SSIM scores exceeding 0.98. These results outperform WAM and are only marginally lower than TrustMark and Robust-Wide. More importantly, under this high-fidelity setting, both variants of MaskWM still maintain near 100% bit accuracy, even under various valuemetric and geometric distortions. This demonstrates significantly better robustness compared to both global and local watermarking baselines. Notably, geometric distortions, which often break existing global watermarking methods, are effectively handled by our robust and reliable MaskWM framework.

Table 1: Comparison with baseline watermarking methods in terms of global watermarking. The **best** and the second best results are highlighted in bold and underlined, respectively.

| Method | Bit Length | PSNR ↑ | SSIM ↑ | No Distortion ↑ | Distortions | |
| | | | | | Valuemetric ↑ | Geometric ↑ |
| --- | --- | --- | --- | --- | --- | --- |
| *Global Watermarking Methods* | | | | | | |
| StegaStamp [22] | 100 | 28.87 | 0.9019 | 0.9990 | 0.9976 | 0.6646 |
| SepMark [24] | 30 | 35.73 | 0.9876 | 0.9957 | 0.9643 | 0.5086 |
| TrustMark [2] | 100 | 41.19 | 0.9922 | 0.9996 | 0.9955 | 0.7868 |
| EditGuard-C [30] | 64 | 37.27 | 0.9332 | 0.9991 | 0.5482 | 0.4925 |
| EditGuard-D [30] | 64 | 32.30 | 0.8199 | 0.9999 | 0.5444 | 0.4975 |
| Robust-Wide [8] | 64 | **41.58** | **0.9923** | **1.0000** | 0.9944 | 0.4951 |
| VINE [14] | 100 | 36.04 | 0.9874 | 0.9997 | 0.9986 | 0.5012 |
| *Local Watermarking Methods* | | | | | | |
| WAM [20] | 32 | 39.32 | 0.9791 | **1.0000** | 0.9986 | 0.8979 |
| MaskWM-D (Ours) | 32 | 39.55 | 0.9814 | **1.0000** | **1.0000** | 0.9998 |
| MaskWM-ED (Ours) | 32 | 39.52 | 0.9828 | **1.0000** | **1.0000** | **1.0000** |

**Local Watermarking Results.** The local watermarking results for all methods are presented in Figure 3. Global watermarking methods suffer a significant drop in extraction accuracy as the watermarked region shrinks, revealing their weakness in local watermarking tasks. In contrast, both WAM and MaskWM maintain high accuracy even with small watermarked areas, and MaskWM consistently outperforms WAM, especially under distortions, demonstrating greater robustness. Between our variants, MaskWM-ED performs better when the watermarked area is small, with the performance gap narrowing as the watermarked area increases.

**Visualized Results.** The visualized watermark patterns embedded by MaskWM-D and MaskWM-ED are shown in Figure 15 in Appendix D.6.

## 4.3 Watermark Localization Comparison

**Settings.** We further compare MaskWM with EditGuard [30] and WAM [20], two methods with watermark localization capabilities. Localization performance is evaluated using the local watermarking dataset described in Sec. 4.2, with Intersection-over-Union (IoU) metrics computed between the predicted and ground-truth regions for both the watermarked and unwatermarked areas.

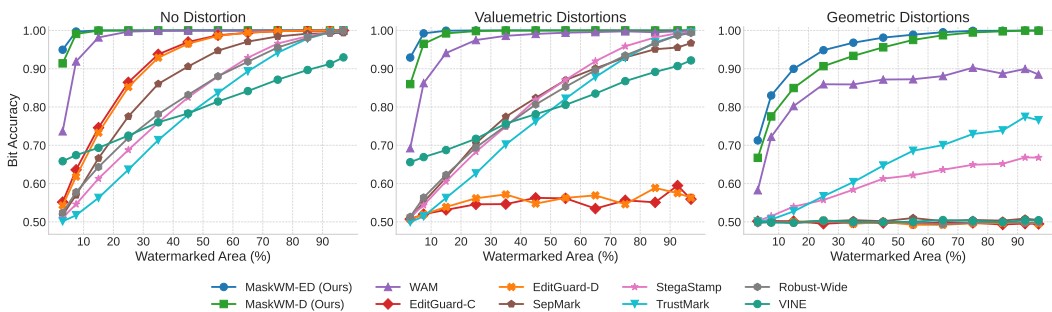

Figure 3: Watermark extraction performance of different methods under different ratios of watermarked areas. The intervals of ratios are: 1-5%, 5-10%, ..., 95-99%, 99-100%. We select the average value for each interval's ratios to stand for the interval (e.g., 3% for 1-5%).

**Results.** The localization results of different methods are presented in Figure 4. First, MaskWM consistently achieves the best localization performance across nearly all watermark ratios and distortion conditions, with WAM showing a noticeable performance gap and EditGuard performing significantly worse. Second, MaskWM-ED outperforms MaskWM-D in localizing small regions, whether watermarked or unwatermarked, especially under distortion conditions. Interestingly, EditGuard-C shows an unusual rise in localization performance for unwatermarked regions as the watermark area increases under non-distorted conditions, a behavior not observed in EditGuard-D, possibly due to overfitting from being trained solely on clean data. The visualized localization results of different methods are provided in Appendix D.7.

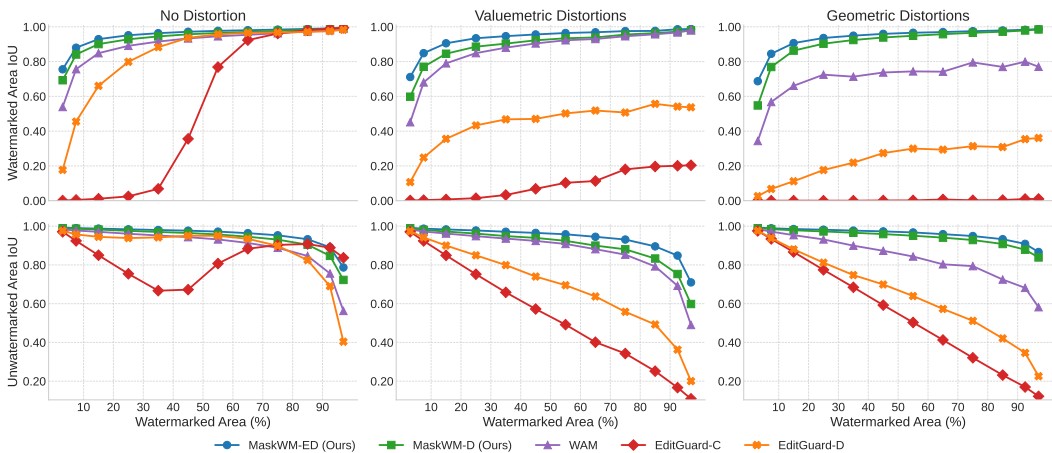

Figure 4: Localization performance of different methods under different ratios of watermarked areas.

## 4.4 Performance Comparisons of Embedding Multiple Watermarks

**Settings.** We further compare MaskWM-ED with WAM [20] under a multi-watermark setting by embedding up to five distinct watermarks into separate masked regions. The evaluation considers extraction accuracy and localization performance under a stricter masking constraint (5% area per region). We report the average watermark extraction accuracy across all embedded watermarks and the mean IoU of the predicted watermark regions. Detailed setup is provided in Appendix C.2.4.

**Results.** Figure 5 presents the comparison results. First, our MaskWM-ED consistently outperforms WAM in both watermark extraction and localization, across all tested numbers of embedded watermarks and under various distortion conditions. Second, despite being trained solely with single-watermark supervision, MaskWM-ED generalizes well to multi-watermark settings, demonstrating strong scalability. However, under geometric distortions, MaskWM-ED's extraction accuracy degrades as the number of embedded watermarks increases. This is mainly due to spatial transfor-

mations (particularly rotation) shrinking the watermark regions in the image corners, reducing the effective area available for extraction (see Figure 20 in Appendix D.8).

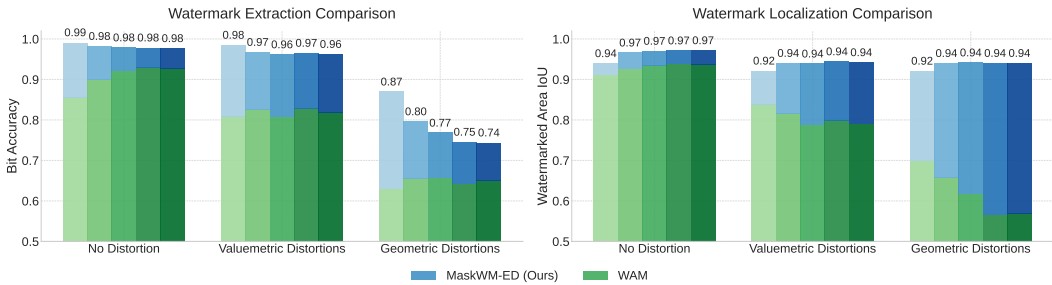

Figure 5: Performance comparison of watermark extraction and localization when embedding multiple watermarks. The bar colors transition from light to dark from left to right, representing the embedding of 1 to 5 different bit strings in a single image.

## 4.5 More Analysis

**Scalability to Different Watermark Bit Lengths.** While WAM is constrained to 32-bit watermarks, our MaskWM readily scales to longer bit lengths. Detailed results are provided in Figure 12 in Appendix D.3. MaskWM maintains high extraction accuracy across various bit lengths. Even at 64 bits, the accuracy experiences only a slight drop and still outperforms WAM at 32 bits. While the accuracy decline becomes more noticeable at 128 bits, MaskWM consistently surpasses WAM under both no distortion and valuemetric distortion conditions. The only exception arises under geometric distortion, where a performance gap emerges when the watermarked region covers between 5% and 75%. These results underscore the scalability and optimization-friendly design of our method, emphasizing its practical advantages.

**Enhancing Robustness against Adaptive Attacks via Fast Fine-tuning.** While MaskWM is trained on a broad range of common distortions, it is impractical to cover all possible scenarios. Fortunately, our framework supports task-specific fine-tuning, allowing it to adapt to new and emerging threats. As a demonstration, we consider adaptive attacks based on variational autoencoders (VAEs) [11], which reconstruct images and may inadvertently remove watermark signals. To counter this, we expand the distortion pool during fine-tuning by incorporating VAE modules from Stable Diffusion v1-4 [17], Bmshj18 [1], and Cheng20 [3]. Training details are provided in Appendix C.1.1. Fine-tuning for 20k steps on a single A6000 GPU (~5 hours) significantly improves robustness against these attacks, as shown in Figure 13 in Appendix D.4, outperforming existing baselines. While there is a slight drop in robustness to other distortions, MaskWM still maintains strong overall performance. These results demonstrate the adaptability of our approach to specific threats.

**Importance of Localization before Extracting Local Watermarks.** We evaluate watermark extraction performance using three masking strategies when processing the watermarked image for extraction: (1) a full mask, which uses the entire image; (2) a predicted mask, which focuses on regions predicted to contain the watermark; and (3) a ground-truth mask, which focuses on the actual watermark-embedded regions. As shown in Figure 14 in Appendix D.5, when watermarks are embedded in small regions, using the full mask significantly reduces accuracy. In contrast, the predicted and ground-truth masks progressively improve performance. This highlights the importance of localizing watermarked regions before extraction to reduce interference from irrelevant content.

**Computation Overhead Evaluation.** We compare training and inference costs of MaskWM with EditGuard [30] and WAM [20], both supporting watermark extraction and localization, summarized in Table 2. MaskWM has fewer parameters than WAM, with similar encoder and decoder sizes; WAM allocates most parameters to the decoder due to its more complex task. In training, MaskWM requires only about 1/15 of WAM's TFLOPs. EditGuard doesn't report training cost, but its data consumption (steps×batch) already reaches 60% of MaskWM 's, despite using fewer strong distortions for robustness. Fewer parameters don't always mean lower inference overhead, as different image processing methods affect memory and speed. Although EditGuard has the fewest parameters, it uses

more memory and runs slower due to embedding both localization and copyright watermarks. In contrast, MaskWM and WAM have lower latency with similar memory use.

Table 2: Training and inference costs of different methods. The inference time is evaluated on a single A6000 with a batch size of 1 by averaging the total processing time over 1000 images.

| Method | # Params (M) | | | Train | | | | | | Inference | | | |
|---|---|---|---|---|---|---|---|---|---|---|---|---|---|
| | Enc | Dec | Total | Steps | Batch | GPU | Time | TFLOPs | Memory | Enc Time | Dec Time | Total Time | Memory |
| EditGuard [30] | 3.6 | 2.6 | 6.2 | 250K | 4 | - | - | - | - | 0.074 s | 0.080 s | 0.154 s | 2.15 GB |
| WAM [20] | 1.1 | 96.0 | 97.1 | 3,680K | 16 | 8 V100 | 90 h | $1.13 \times 10^{16}$ | - | 0.015 s | 0.017 s | 0.032 s | 2.33 GB |
| MaskWM (Ours) | 31.1 | 32.2 | 63.3 | 100K | 16 | 1 A6000 | 20 h | $7.74 \times 10^{14}$ | 25.84 GB | 0.009 s | 0.022 s | 0.031 s | 2.21 GB |

## 5  Conclusion

In this paper, we propose MaskWM, a simple and efficient framework for both global and local image watermarking. Its core design introduces masks in encoding and decoding stages to guide the encoder and decoder to learn local embedding and extraction. Extensive experiments demonstrate MaskWM 's superior performance in local watermark extraction and localization, along with high efficiency and adaptability. We hope our simple design can inspire future research to further enhance the practicality and functionality of watermarking models.

#### Acknowledgments

This research / project is supported by the National Research Foundation, Singapore and Infocomm Media Development Authority under its Trust Tech Funding Initiative. Any opinions, findings and conclusions or recommendations expressed in this material are those of the author(s) and do not reflect the views of National Research Foundation, Singapore, and Infocomm Media Development Authority.

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

# A Limitations

Despite the relatively high PSNR and SSIM of the watermarked images produced by our method, noticeable artifacts are still present in some cases. To address this issue, we explored multiple approaches and ultimately adopted a JND-based method [23] to effectively suppress artifacts (as shown in Appendix D.1). Nevertheless, some images still exhibit visible artifacts after watermark embedding, particularly in smooth background regions. This issue is also present in WAM [20] and reflects a common limitation in local watermarking methods, possibly arising from the inherent trade-off between imperceptibility and local robustness. Our focus in this work is to propose a watermarking framework centered on a mask mechanism, rather than to exhaustively pursue visual quality improvements. We have not further explored advanced strategies such as improved loss functions, auxiliary modules for visual enhancement, or more effective algorithms for watermark strength modulation. These aspects are left for future work to enhance the practicality and visual fidelity of our method.

# B Impact Statement

We propose a simple, efficient, and flexible image watermarking framework whose core component, a masking mechanism, can be easily integrated into existing watermarking models. This mechanism enhances both the functionality and robustness of watermarking systems, significantly improving their practicality in real-world scenarios. The support for local watermark extraction allows reliable recovery of watermark information even when large portions of the image are tampered with. This is particularly valuable for forensic verification in areas such as misinformation mitigation and evidence integrity analysis. The ability to localize watermark regions enables systems to identify precisely which parts of an image contain embedded signals, facilitating transparent content attribution and enabling fair decision-making in legal and copyright-related contexts. Furthermore, local watermark embedding allows protection of specific regions within an image, which is useful when only certain parts require ownership marking or when content from multiple sources needs independent tracking and licensing. By improving spatial controllability while maintaining low computational cost and compatibility with existing architectures, our approach makes robust and fine-grained watermarking more accessible and deployable across diverse applications.

# C More Details

## C.1 Training

### C.1.1 Details

All images are resized and center-cropped to $256 \times 256$ during training. Training is conducted for 100k steps with a batch size of 16 on a single NVIDIA A6000 GPU. We use the AdamW optimizer with a learning rate of $1 \times 10^{-4}$, and apply a cosine learning rate scheduler with 2k warm-up steps. We adopt an easy-to-hard training strategy inspired by TrustMark [2]. During the first 0.5k steps, the mask is set to full (i.e., all ones) and no distortion is applied. From step 0.5k to 1k, we introduce all types of masks. After 1k steps, distortions are added. The encoder loss weight $\beta_{\text{enc}}$ is fixed at 1, while the decoder loss weight $\beta_{\text{dec}}$ is initially set to 20 and linearly decayed to 0.2 over the first 5k steps. The mask loss weight $\alpha$ is set to 0.5. The JND module in the encoder is introduced and tuned starting from step 5k, with the scaling factor $\mu$ set to 1.

During fine-tuning with adaptive attacks, VAE-based distortions are applied with a 50% probability, while the original distortion types are retained for the remaining 50%. The hyperparameters are set as follows: $\beta_{\text{dec}} = 0.3$, learning rate $= 1 \times 10^{-4}$, and the quality levels for *Bmshj18* and *Cheng20* are both set to 5.

### C.1.2 Distortions

**Valuemetric Distortions.** During training, valuemetric robustness is enhanced by randomly sampling from a set of ten common distortions: JPEG Compression, Gaussian Filter, Gaussian Noise, Median Filter, Salt&Pepper Noise, Resize, Brightness, Contrast, Hue, and Saturation. The distortion parameters are set as follows:

- **JPEG Compression:** quality factor = 50.

- **Gaussian Filter:** kernel size = 1, sigma = 5.

- **Gaussian Noise:** mean = 0, standard deviation = 0.1.

- **Median Filter:** kernel size = 5.

- **Salt&Pepper Noise:** noise ratio = 0.1.

**Geometric Distortions.** To improve geometric robustness, we randomly sample from three typical geometric transformations: Rotation, Perspective, and Horizontal Flip. The specific configurations are:

- **Rotation:** angle sampled from $[-90°, 90°]$.

- **Perspective:** distortion scale sampled from $[0.1, 0.5]$.

- **Horizontal Flip:** no parameters.

## C.2 Evaluation

### C.2.1 Resolution Scaling

Algorithm 1 is adapted from the resolution scaling method described in TrustMark [2]. This algorithm enables a watermark model trained on images with a fixed resolution to be applied at arbitrary resolutions without sacrificing performance.

---

**Algorithm 1:** Resolution scaling - watermark embedding on arbitrary resolution images

---

**Input:** Original image $\mathbf{x}$, [binary watermark vector $\mathbf{w}$]
**Output:** Watermarked image $\mathbf{y}$
**Data:** Embedding network $\mathbf{E}$ trained on the resolution of $m \times n$

1 $H, W \leftarrow \mathbf{x}.\text{height}, \mathbf{x}.\text{width}$
2 $\mathbf{x} \leftarrow \mathbf{x}/127.5 - 1$ ;                              // Normalize to range [-1, 1]
3 $\bar{\mathbf{x}} \leftarrow \text{interpolate}(\mathbf{x}, (m, n))$
4 $\mathbf{r} \leftarrow \mathbf{E}(\bar{\mathbf{x}}, \mathbf{w}) - \bar{\mathbf{x}}$ ;                              // residual image
5 $\mathbf{r} \leftarrow \text{interpolate}(\mathbf{r}, (H, W))$
6 $\mathbf{y} \leftarrow \text{clamp}(\mathbf{x} + \mathbf{r}, -1, 1)$
7 $\mathbf{y} \leftarrow \mathbf{y} \times 127.5 + 127.5$

---

### C.2.2 Distortions

**Valuemetric Distortions.** We apply ten types of valuemetric distortion with the following parameter settings to evaluate robustness:

- **JPEG Compression:** quality factor = 60.

- **Gaussian Filter:** kernel size = 1, sigma = 3.

- **Gaussian Noise:** mean = 0, standard deviation = 0.05.

- **Median Filter:** kernel size = 3.

- **Salt&Pepper Noise:** noise ratio = 0.05.

- **Resize:** scaling factor = 0.5.

- **Brightness Adjustment:** range $(0.7, 1.3)$.

- **Contrast Adjustment:** range $(0.7, 1.3)$.

- **Hue Adjustment:** range $(-0.1, 0.1)$.

- **Saturation Adjustment:** range $(0.7, 1.3)$.

**Geometric Distortions**  We apply three types of geometric distortion with the following parameter settings to evaluate robustness:

- **Rotation:** angle sampled from $[-30°, 30°]$.
- **Perspective:** distortion scale sampled from $[0.1, 0.3]$.
- **Horizontal Flip:** no parameters.

### C.2.3 Global and Local Watermarking Comparison

**Evaluation Setup.**  For global watermarking, we sample 1,000 images from the MS-COCO 2014 validation set. We report PSNR and SSIM to assess visual quality, and Bit Accuracy to measure watermark extraction performance. For local watermarking, we construct a comprehensive evaluation set from all 41,000 images in the validation split. We divide the dataset into 12 subsets based on the ratio of masked area to the full image: 1–5%, 5–10%, 10–20%, ..., 80–90%, 90–95%, and 95–99%. From each subset, 400 images are randomly selected. To simulate both inpainting and outpainting scenarios, we also include inverted masks, i.e., if the original mask covers $a$–$b$% of the image, the inverted mask covers $(100-b)$–$(100-a)$%. This yields 800 image-mask pairs per subset and a total of 9,600 for evaluation.

**Embedding Strategy.**  Baselines and MaskWM-D embed watermarks across the entire image, but the unmasked regions are replaced with the original image to localize the watermark. MaskWM-ED embeds watermark bits only within the masked region and similarly restores the unmasked parts, ensuring true region-specific watermarking.

### C.2.4 Multi-Watermark Embedding Setup

Figure 6 illustrates the spatial arrangement of the five non-overlapping masked regions used in our multi-watermark experiments. These regions are fixed at the center, top-left, top-right, bottom-left, and bottom-right of the image, forming a checkerboard-like layout when all five are active. Unlike WAM, which allocates 10% of the image area to each region, we restrict each region to 5%, making the watermark extraction task more challenging. For each configuration (1 to 5 watermarks), we randomly sample 400 images from the MS-COCO 2014 validation set. During evaluation, we use OpenCV's `cv2.connectedComponents` to segment the predicted mask into disjoint components, enabling region-wise watermark extraction and scoring.

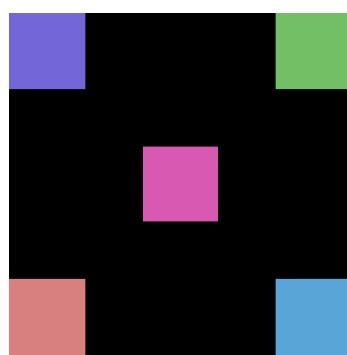

Figure 6: Checkerboard-like arrangement of masked regions for multi-watermark embedding. Different colors indicate regions where different watermarks bits are embedded, while black regions contain no embedded information.

## D  More Results

### D.1 Effects of Different Visual Quality Enhancement Methods

Our experiments demonstrate that using only an MSE loss in the pixel space to constrain the difference between the watermarked and original images (referred to as the Base version) already yields high PSNR and SSIM scores. However, despite the favorable metrics, the resulting images often contain visible artifacts that may compromise perceived visual quality. To address this issue, we investigate several enhancement strategies: (1) incorporating a GAN [9] loss, (2) adding a perceptual constraint via LPIPS [29] loss in the feature space, and (3) modulating the watermark signal using a Just Noticeable Difference (JND) [23] module. We conduct experiments on the MaskWM-D variant, but the findings similarly hold for MaskWM-ED.

As shown in Figure 7, when the weights of the GAN and LPIPS losses are low, they fail to effectively suppress artifacts. Increasing these weights reduces artifacts but adversely affects the performance

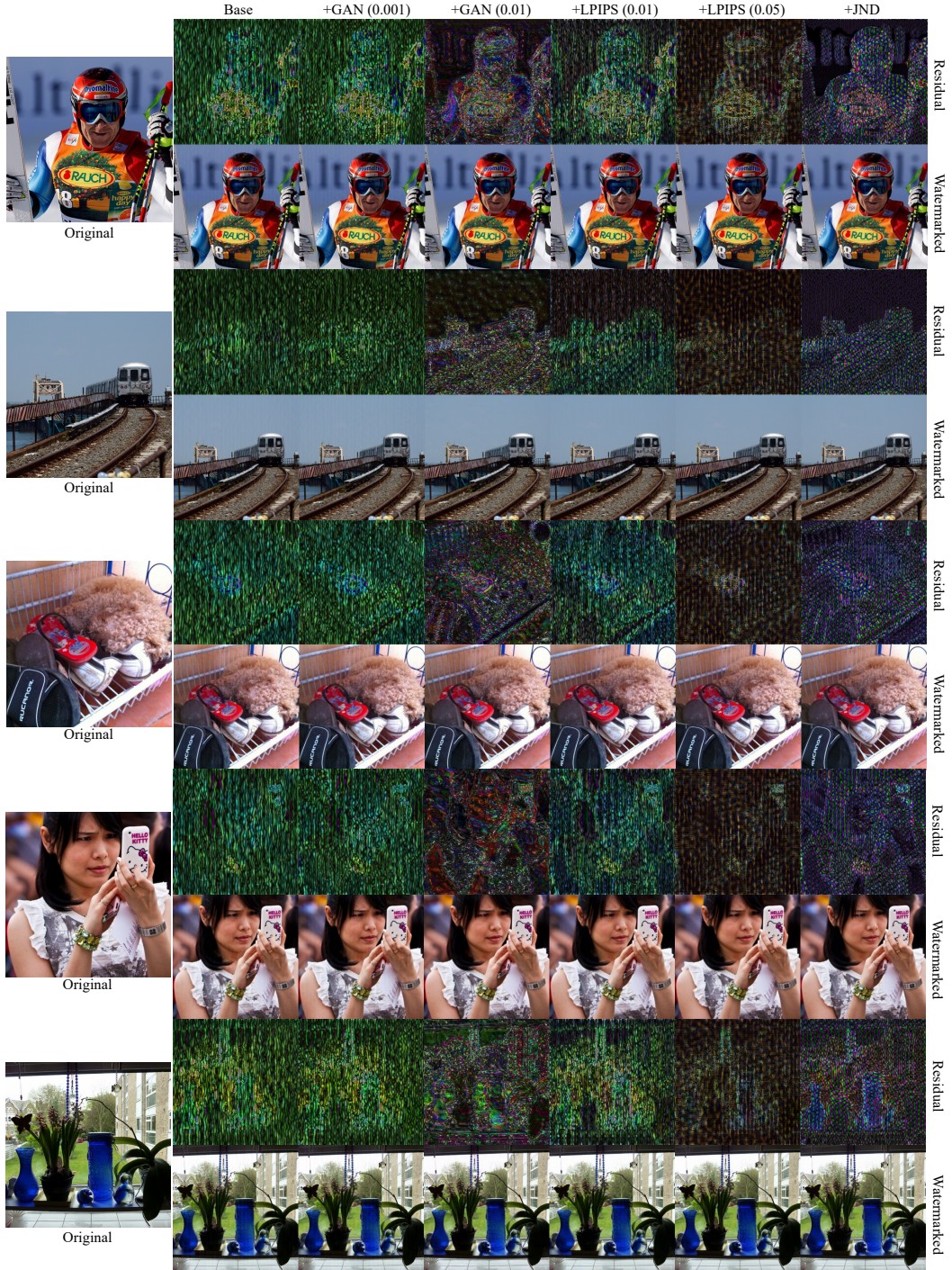

Figure 7: Visualization results of watermark using different visual quality enhancement methods. Zoom in to see more details. The numbers in parentheses indicate the corresponding loss weights. For example, GAN (0.001) means the GAN loss is assigned a weight of 0.001 in the total loss function, i.e., $\mathcal{L}_{\text{total}} = \beta_{\text{enc}}\mathcal{L}_{\text{enc}} + \beta_{\text{dec}}\mathcal{L}_{\text{dec}} + 0.001\mathcal{L}_{\text{GAN}}$, and LPIPS (0.01) indicates that the LPIPS loss is weighted by 0.01, i.e., $\mathcal{L}_{\text{enc}} = \mathcal{L}_{\text{MSE}}(I_{wm}, I_{orig}) + 0.01\mathcal{L}_{\text{LPIPS}}$. To ensure a fair comparison, we adjust either the strength of the added watermark residual or the JND modulation coefficient to maintain comparable PSNR and SSIM across different settings.

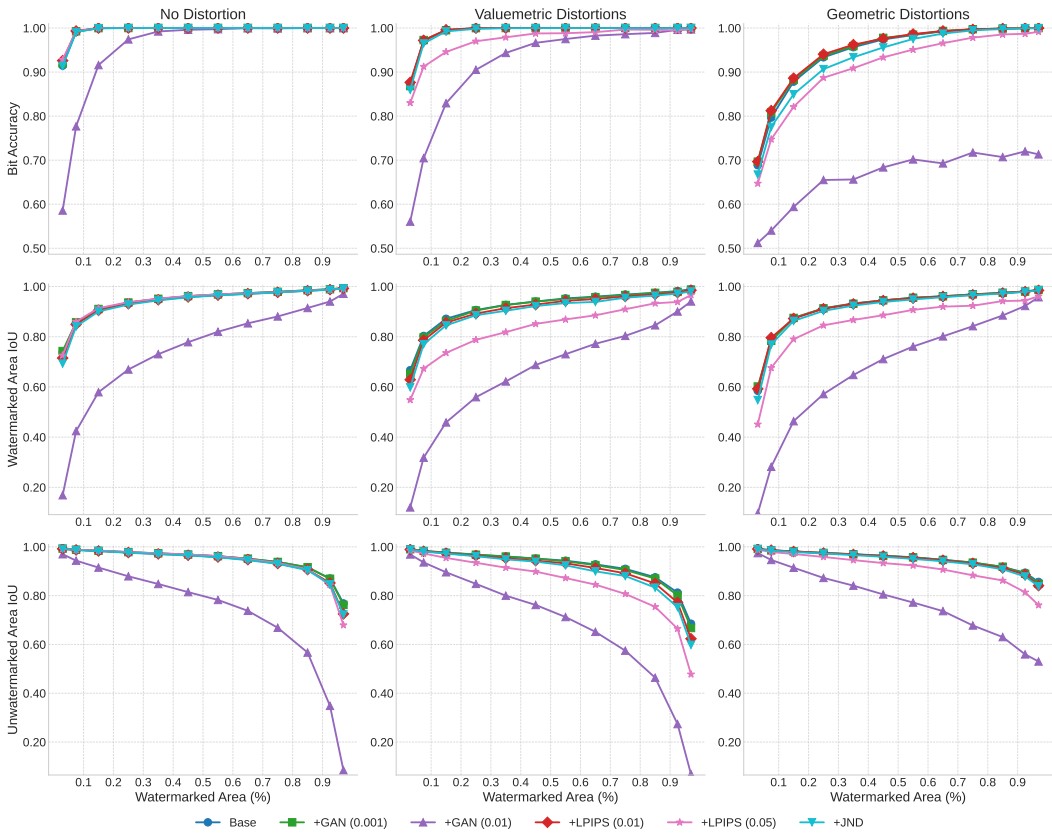

Figure 8: Local watermark extraction and localization performance of MaskWM-D trained with different visual quality enhancement methods.

of local watermark extraction and localization, as illustrated in Figure 8. In contrast, applying JND-based modulation proves more effective: it significantly reduces visible artifacts while maintaining performance comparable to the Base version. These findings suggest that, unlike global perceptual losses such as GAN or LPIPS, JND offers a more adaptive and content-aware modulation strategy. It effectively suppresses visual artifacts while preserving the watermark's integrity, making it a practical choice for visual quality enhancement.

### D.2 Results under Specific Distortions

See Figure 9, Figure 10, Figure 11.

### D.3 Scalability to Different Watermark Bit Lengths

See Figure 12.

### D.4 Enhancing Robustness against Adaptive Attacks via Fast Fine-tuning

See Figure 13.

### D.5 Importance of Localization before Extracting Local Watermarks

See Figure 14.

### D.6 Visualization Results of Global and Local Watermark Embedding

See Figure 15.

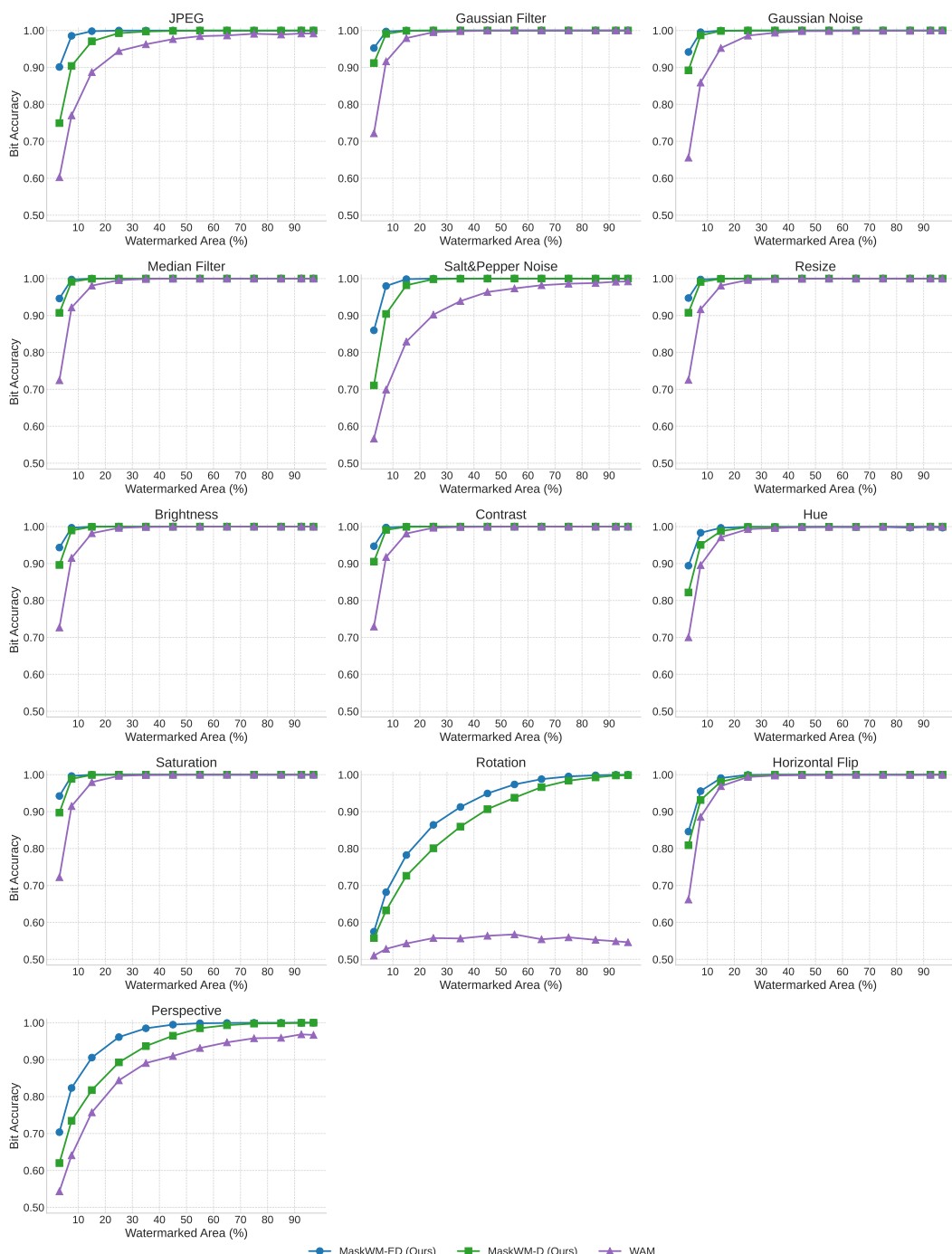

Figure 9: Watermark extraction performance under various specific distortions.

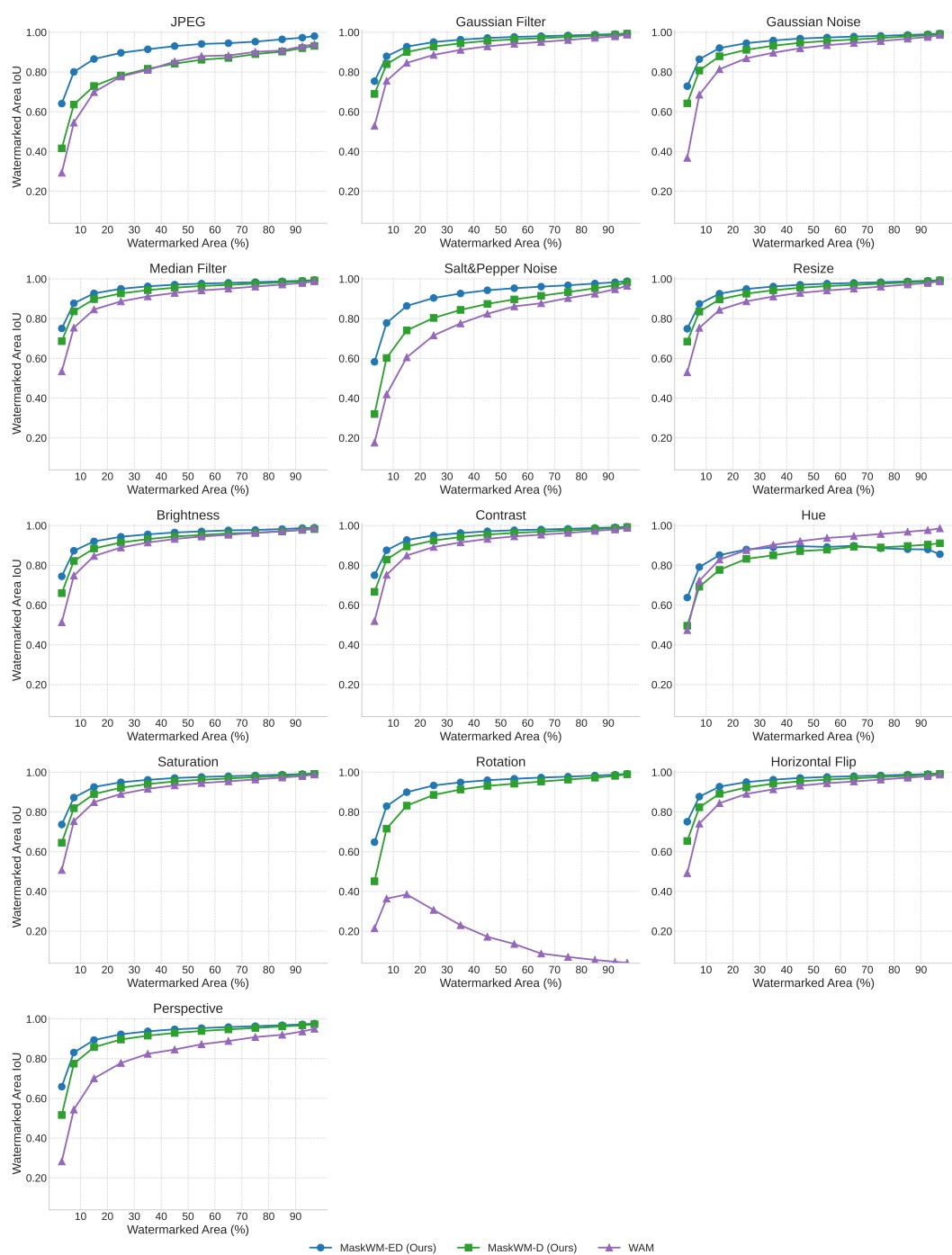

Figure 10: Localization performance of the watermarked area under various specific distortions.

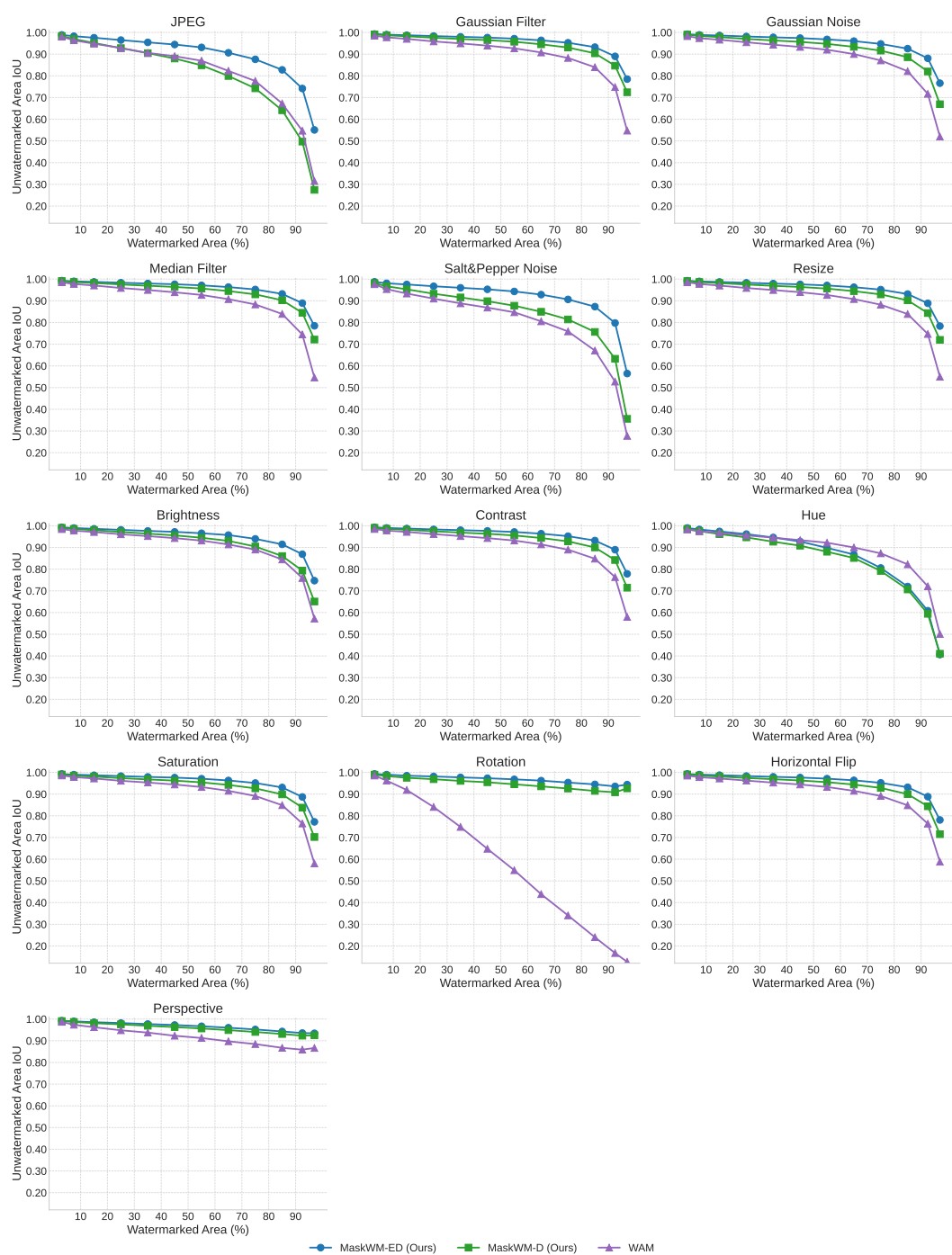

Figure 11: Localization performance of the unwatermarked area under various specific distortions.

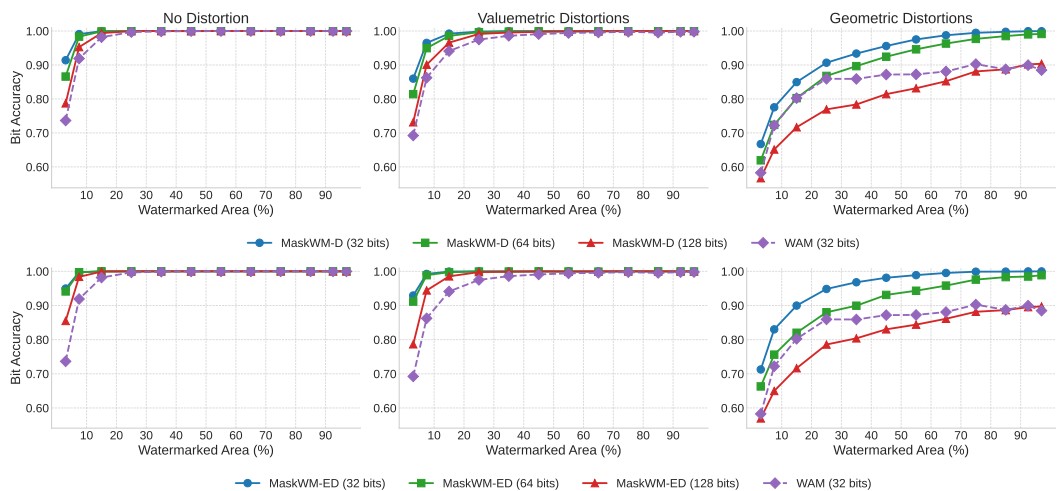

Figure 12: Watermark extraction performance of MaskWM-D and MaskWM-ED with different bits length. We also show the results of WAM for comparison.

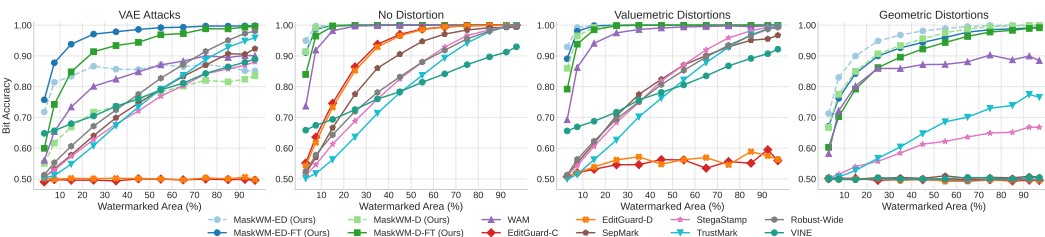

Figure 13: The effect of VAE fine-tuning on the robustness of MaskWM. Fine-tuning the VAE enhances robustness against VAE attacks, with minimal impact on the original robustness performance.

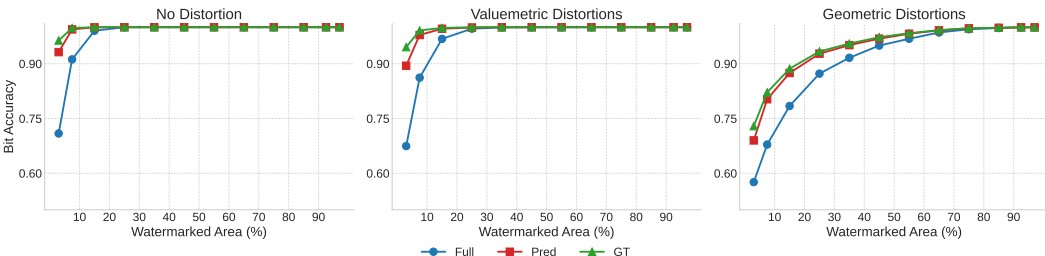

Figure 14: Watermark extraction performance using different masking strategies during decoding.

## D.7 Visualization Results of Localization

See Figure 16, Figure 17, Figure 18, Figure 19.

## D.8 Visualization Results of Multiple Watermarks Embedding

See Figure 20.

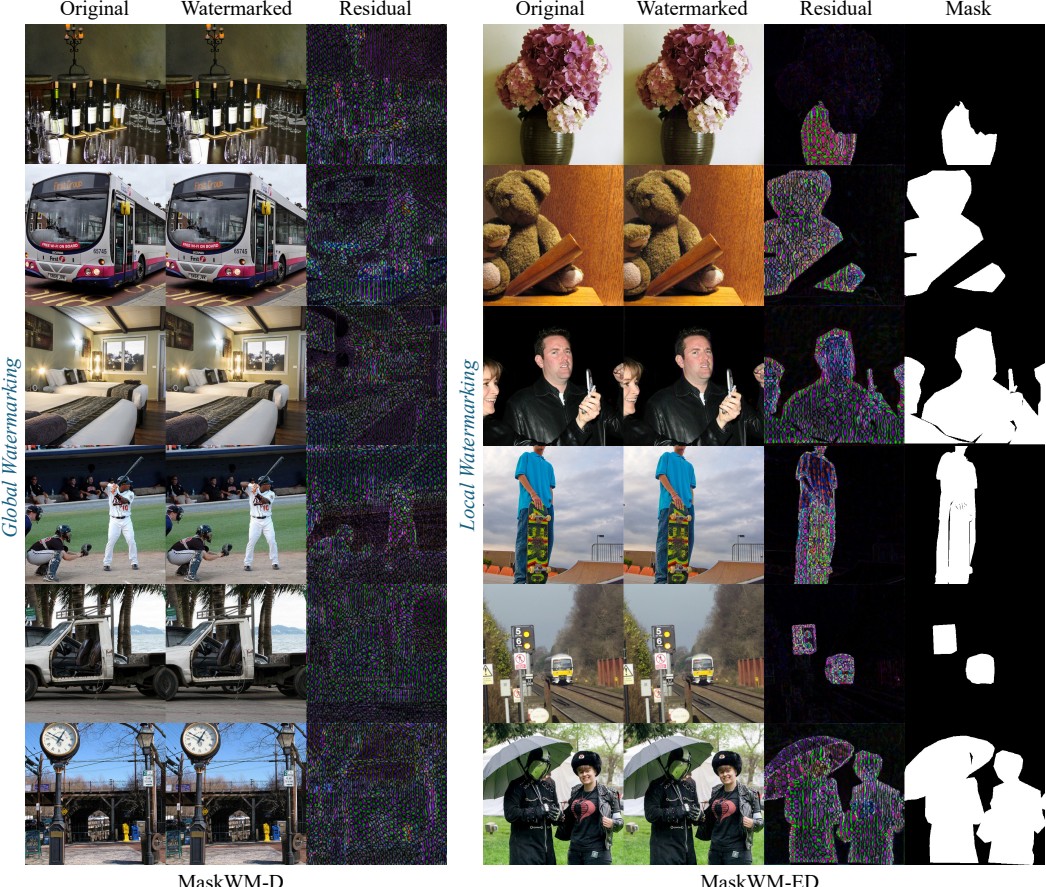

Figure 15: Visualization results of global watermark embedding using MaskWM-D and local watermark embedding using MaskWM-ED. The residual image is acquired by $10 \times |I_{wm} - I_{orig}|$ for observation, highlighting the residual more clearly. The same residual visualization strategy is applied in the following figures as well.

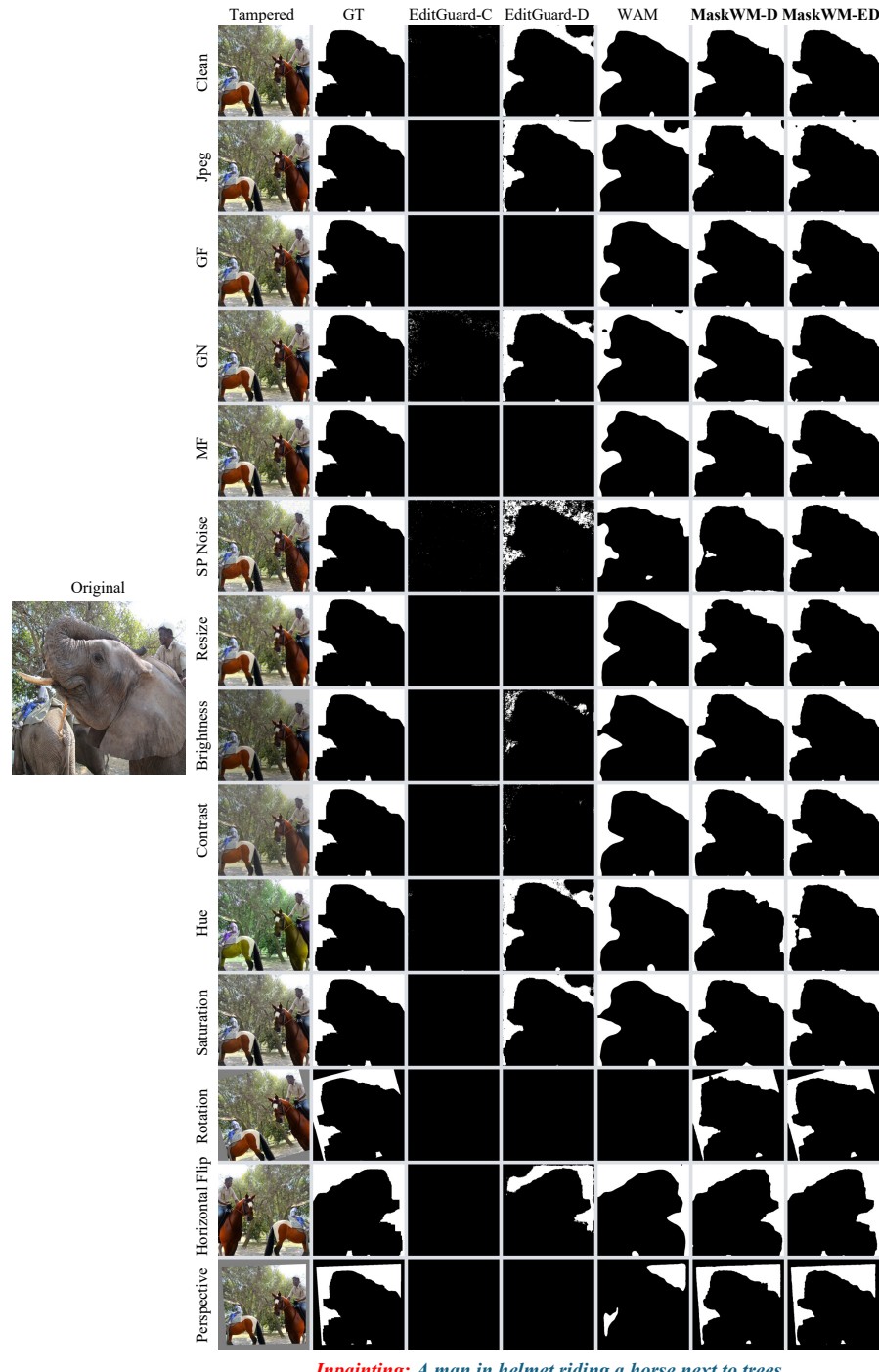

Figure 16: Visualization results of watermark localization using different methods. The inpainting results are obtained by applying *stable-diffusion-2-inpainting* [17] to the masked regions for content reconstruction.

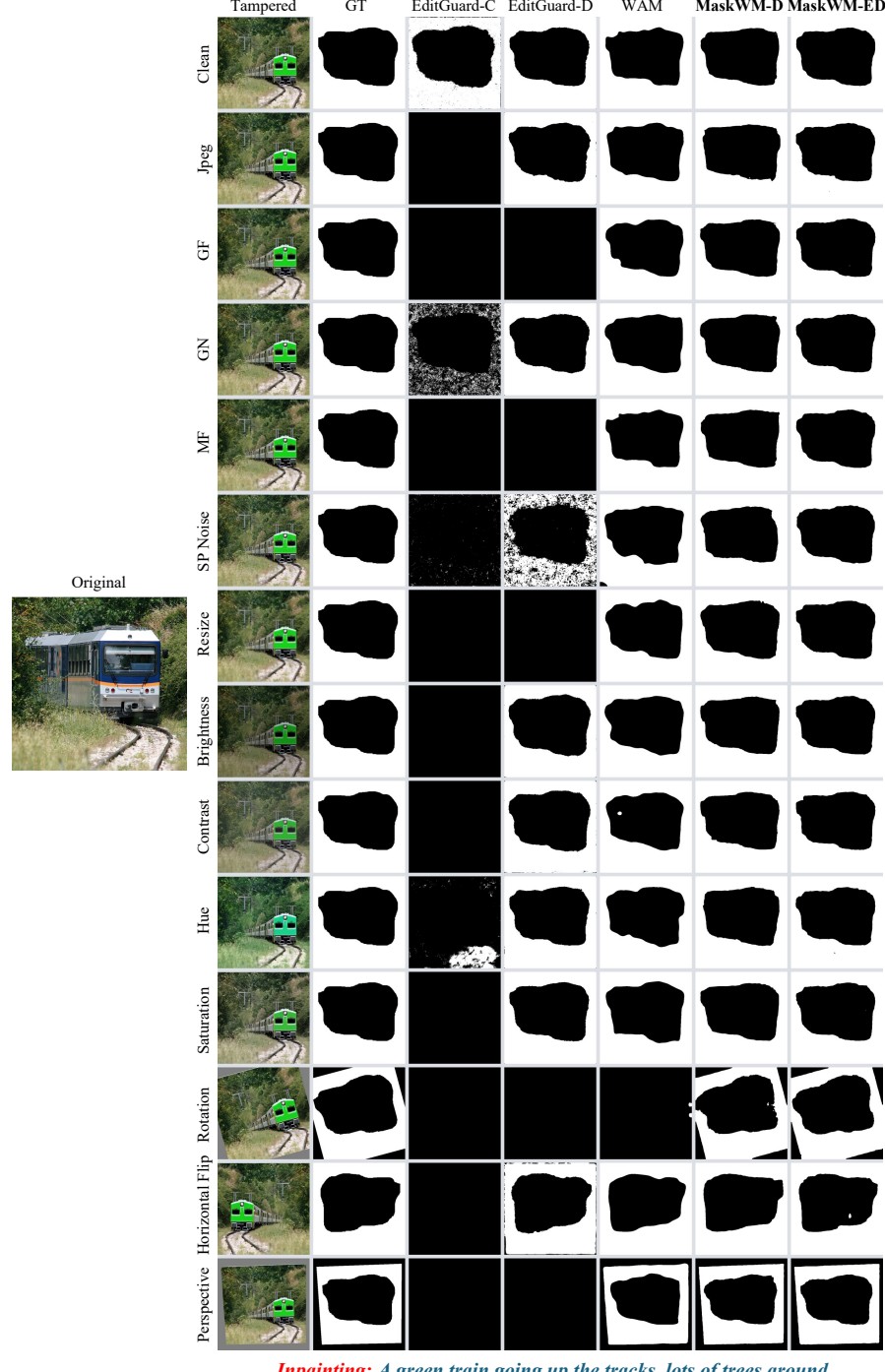

Figure 17: Visualization results of watermark localization using different methods. The inpainting results are obtained by applying *stable-diffusion-2-inpainting* [17] to the masked regions for content reconstruction.

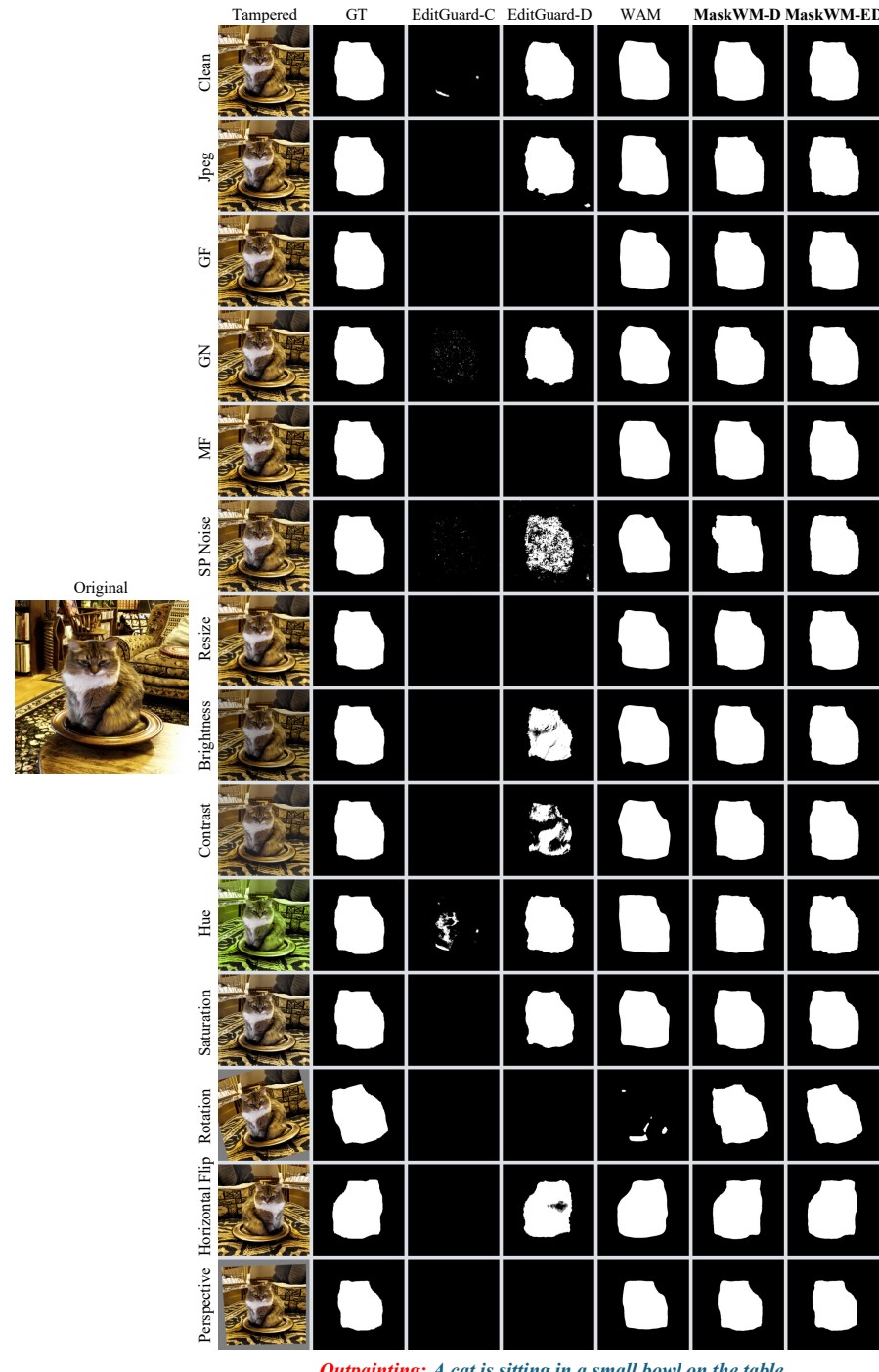

*Outpainting: A cat is sitting in a small bowl on the table.*

Figure 18: Visualization results of watermark localization using different methods. The outpainting results are obtained by applying *stable-diffusion-2-inpainting* [17] to the masked regions for content reconstruction.

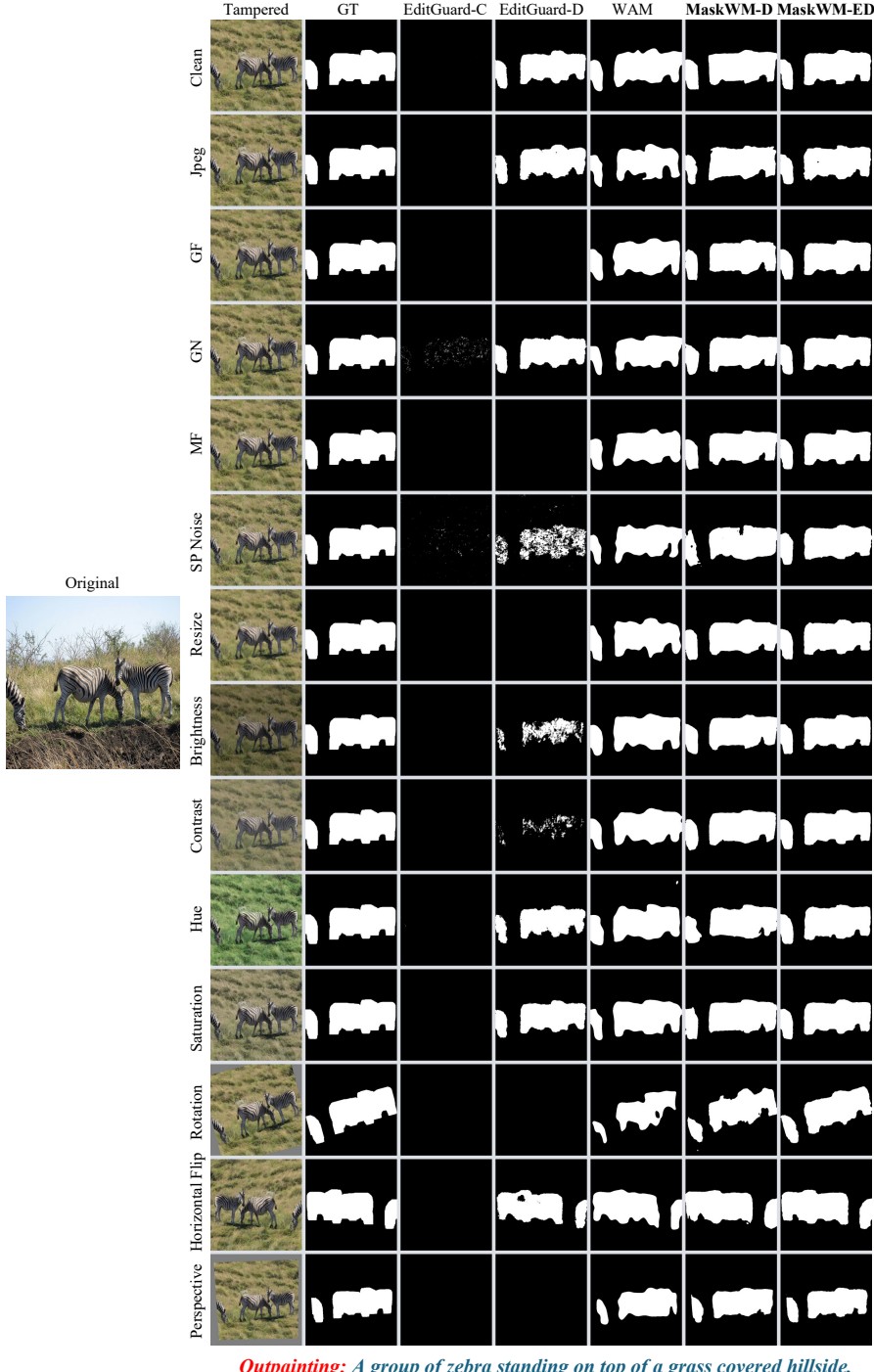

*Outpainting: A group of zebra standing on top of a grass covered hillside.*

Figure 19: Visualization results of watermark localization using different methods. The outpainting results are obtained by applying *stable-diffusion-2-inpainting* [17] to the masked regions for content reconstruction.

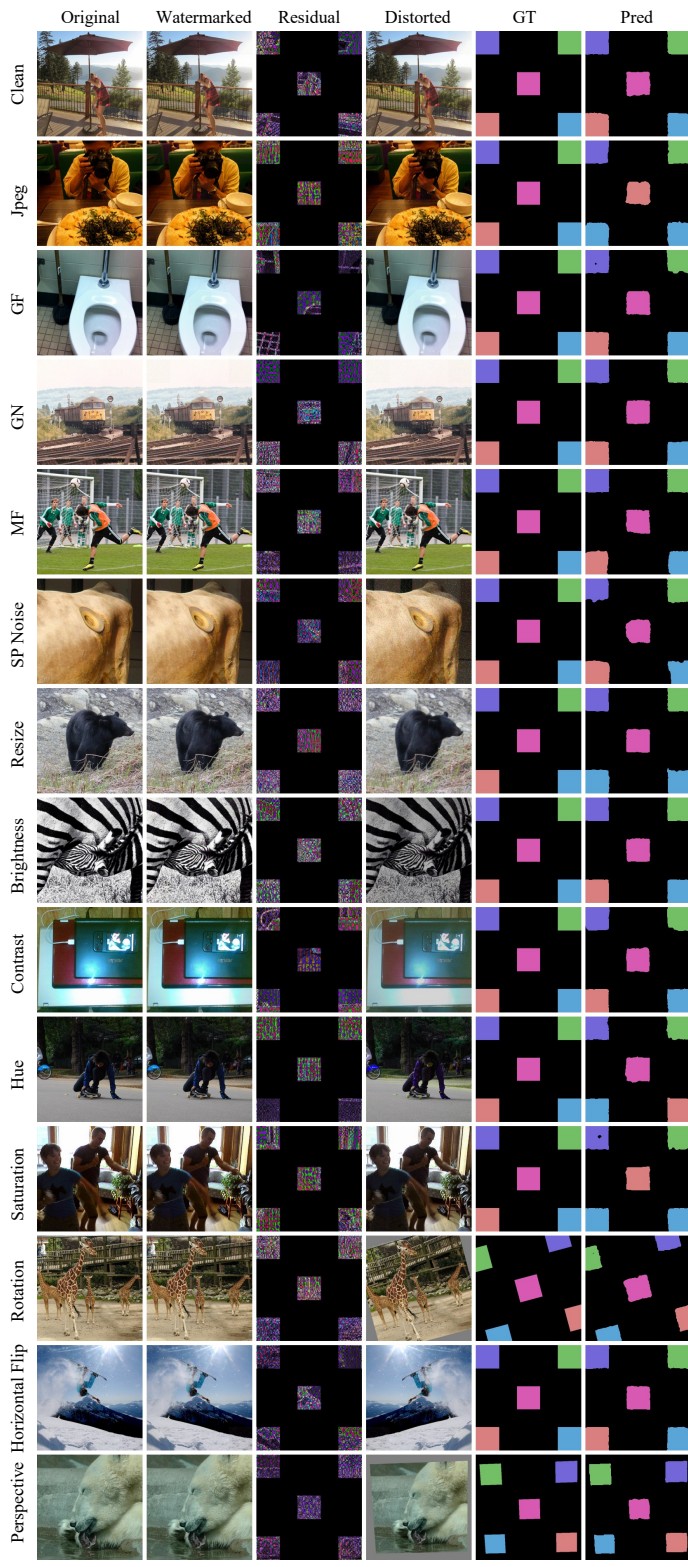

Figure 20: Visualization of multi-watermark embedding and localization results. In the GT column, different colors in the mask indicate different watermark messages.

