# OpenReview forum: "Mask Image Watermarking"
_NeurIPS.cc/2025/Conference — NeurIPS 2025 poster_

### Official Review · Reviewer_fV1g · 2025-06-24

**Clarity:** 3
**Significance:** 2
**Originality:** 2
**Rating:** 4
**Confidence:** 3

**Summary:**

This paper introduces MaskMark, a framework for image watermarking with two variants: MaskMark-D for global/local watermark extraction and tamper detection, and MaskMark-ED for fine-grained local watermarking with enhanced robustness. The method leverages a masking mechanism during training to enable flexible watermark localization and extraction, achieving state-of-the-art performance in accuracy, efficiency, and adaptability while preserving image quality.

**Questions:**

Elaborate the genuine technical novelties. Explicitly characterize the scenarios where the masking strategy provides unique advantages beyond what could be achieved through existing robustness techniques.

The paper shows impressive results against localized attacks, but real-world adversaries often combine multiple attack types. Does MaskMark maintain its superiority under such composite attacks, and does the proposed architecture have inherent limitations against certain attack combinations?

**Ethical Concerns:**

["NO or VERY MINOR ethics concerns only"]

**Final Justification:**

Based on the original manuscript and the authors' responses, I have no major concerns regarding this paper.

**Limitations:**

yes

**Paper Formatting Concerns:**

No major formatting issues found.

**Quality:**

3

**Strengths And Weaknesses:**

While the paper correctly identifies meaningful limitations in current watermarking techniques, the proposed solutions represent relatively incremental improvements over existing approaches. For instance, the core masking mechanism in MaskMark shares conceptual similarities with prior work on robustness against random cropping (Ref. [30]), essentially adapting established training paradigms through modified masking strategies rather than introducing fundamentally novel architectures or methodologies.

---

> ### Author Rebuttal · Authors · 2025-07-31
>
> # Author Response (Reviewer fV1g)
> Dear Reviewer, thank you very much for your careful review of our paper and thoughtful comments. We hope the following responses can help address your concerns.
>
> ___
>
> **Q1** Elaborate the genuine technical novelties. Explicitly characterize the scenarios where the masking strategy provides unique advantages beyond what could be achieved through existing robustness techniques.
>
> **A1:** Thank you for raising this important point. Below, we elaborate on the genuine technical novelties introduced by MaskMark and characterize the unique scenarios where our masking strategy provides clear advantages over prior watermarking techniques and robustness mechanisms.
>
> **Functional Innovation: Native Local Watermark Embedding.**
> - MaskMark is, to our knowledge, the **first method to introduce a spatial mask during the encoding phase to enable native local watermark embedding**.
> - While WAM and other global watermarking methods can embed a global watermark and then applies post-hoc cropping to simulate local watermark embedding (which introduces **signal loss and robustness issues**), MaskMark embeds the watermark directly into designated regions by conditioning encoder on a spatial mask. This results in **improved robustness and precise regional protection**.
>
> **Methodological Innovation: A Unified Mask Mechanism Enabling Multiple Functions.**
> - Beyond native local embedding, the introduced mask mechanism provides **a unified design that enables multiple watermarking functionalities**: local watermark extraction, watermark localization and multi-watermark embedding and extraction.
> - While existing methods may achieve some of these functions individually, they typically rely on **task-specific architectures or ad-hoc procedures**, and **cannot achieve all three simultaneously within a single coherent framework**. Moreover, their performance or efficiency also tends to fall short compared to ours. Below, we explain the advantages of our approach.
>     - **Compared to Global Watermarking Methods that Use Crop as a Distortion.** Some global watermarking methods (e.g., HiDDeN, MBRS) introduce Crop as a distortion during training to improve robustness. However, our mask mechanism differs fundamentally in both its role and effectiveness:
>         - **Persistent Throughout Encoding and Decoding:** The mask is not just a training-time augmentation but a central component in both the encoder and decoder during training and inference. This persistent guidance enables the model to learn true spatial awareness and region-specific signal embedding/extraction.
>         - **Supports Learnable “Locate-then-Extract” Decoding:** Our decoder includes a dedicated mask prediction module, allowing the model to first locate the watermark region and then extract the message from that region only. This significantly reduces interference from unmarked areas and improves decoding robustness. We empirically validate this in Sec. 4.5 (“Importance of Localization before Extracting Local Watermarks”).
>         - **Improves Embedding Signal Concentration:** Because the encoder/decoder is always guided by a mask, the watermark signal is learned to be concentrated within the specified region, rather than being diluted across the entire image. In contrast, global methods that see cropped distortions during training still tend to spread the watermark signal across the full image due to the global decoding objective. This limits their true robustness to local extraction, especially when the marked region is small.
>     - **Compared to WAM (Localized Watermarking Method).**
>         - While WAM simultaneously supports local watermark extraction, watermark localization, and multi-watermark embedding and extraction, its design introduces several challenges.
>             - The core idea of WAM is to **treat watermark extraction as a pixel-wise semantic segmentation problem**. Each pixel independently carries bits of the watermark, which are later averaged during inference.
>             - Extracting meaningful information from individual pixels is inherently difficult, leading to **slow convergence, high training complexity, and limited capacity for longer watermark messages** (as noted in the WAM paper).
>             - The final extraction relies on a **non-trainable and naive** voting/averaging mechanism across pixels, which cannot be optimized end-to-end and performs poorly when the watermarked region is small or per-pixel accuracy is low.
>         - In contrast, MaskMark is fundamentally built around spatially-aware encoding and decoding. The encoder and decoder are both guided by a spatial mask, enabling the system to:
>             - Embed and extract information across multiple pixels within a localized region, which **lowers training difficulty and allows extension to longer messages**.
>             - Train all components end-to-end, including both mask prediction and region-wise watermark embedding/extraction, avoiding WAM’s reliance on non-trainable inference heuristics. This leads to **improved robustness in local watermark extraction**
>     - **Compared to EditGuard (Watermark Localization Method).**
>         - EditGuard embeds two separate types of signals: a 1D copyright watermark and a 2D fragile localization watermark to achieve watermark localization. This dual embedding comes with trade-offs:
>             - The addition of a second watermark signal increases visual distortion and complexity.
>             - Ensuring robustness of both the copyright and localization watermarks simultaneously is difficult and may involve conflicting optimization objectives, limiting the achievable performance.
>         - The watermark localized by MaskMark serves as a copyright watermark, which allows robustness and traceability to be ensured simultaneously. This leads to a higher potential performance upper bound and minimal impact on visual quality.
>
> ___
>
> **Q2:** Does MaskMark maintain its superiority under such composite attacks, and does the proposed architecture have inherent limitations against certain attack combinations?
>
> **A2:** We appreciate your question regarding robustness under composite attacks, which indeed reflect more realistic adversarial scenarios.
> - To evaluate this, we conducted extensive experiments where combinations of multiple distortions were applied to the watermarked images.
>     - We considered a total of 13 distortions: 10 valuemetric distortions (JPEG Compression, Gaussian Filter, Gaussian Noise, Median Filter, Salt & Pepper Noise, Resize, Brightness, Contrast, Hue and Saturation) and 3 geometric distortions (Rotation, Perspective and Horizontal Flip).
>     - During evaluation, we randomly selected and applied between 1 to 5 of these distortions to simulate composite attacks.
> - As shown in Table 1, MaskMark maintains strong performance even as the number of combined distortions increases.
>     - While there is a natural drop in extraction accuracy with more severe corruption, the results remain at a high level across all combinations tested.
>     - Importantly, we did not observe any inherent architectural limitations or specific attack combinations under which MaskMark fails disproportionately, further demonstrating its robustness and practical utility in real-world settings.
>
> **Table 1. Watermark extraction accuracy of MaskMark-D under varying numbers of random combined distortions across different watermarked area ratios.**
> | Number of Random Combined Distortions / Watermarked Area (%) |  1~5  |  5~10 | 10~20 | 20~30 | 30~40 | 40~50 | 50~60 | 60~70 | 70~80 | 80~90 | 90~95 | 95~99 |
> |:---------------------------------------:|:-----:|:-----:|:-----:|:-----:|:-----:|:-----:|:-----:|:-----:|:-----:|:-----:|:-----:|:-----:|
> |                    1                    | 0.803 | 0.907 | 0.946 | 0.969 | 0.981 | 0.989 | 0.992 | 0.995 | 0.998 | 0.998 | 0.999 | 0.999 |
> |                    2                    | 0.710 | 0.799 | 0.869 | 0.910 | 0.916 | 0.931 | 0.944 | 0.953 | 0.962 | 0.965 | 0.967 | 0.974 |
> |                    3                    | 0.621 | 0.707 | 0.772 | 0.825 | 0.846 | 0.861 | 0.877 | 0.891 | 0.899 | 0.895 | 0.898 | 0.903 |
> |                    4                    | 0.573 | 0.644 | 0.686 | 0.729 | 0.756 | 0.769 | 0.785 | 0.802 | 0.815 | 0.816 | 0.855 | 0.833 |
> |                    5                    | 0.538 | 0.575 | 0.617 | 0.654 | 0.666 | 0.697 | 0.706 | 0.712 | 0.730 | 0.728 | 0.739 | 0.754 |
>
> **Table 2. Watermark extraction accuracy of MaskMark-ED under varying numbers of random combined distortions across different watermarked area ratios.**
> | Number of Random Combined Distortions / Watermarked Area (%) |  1~5  |  5~10 | 10~20 | 20~30 | 30~40 | 40~50 | 50~60 | 60~70 | 70~80 | 80~90 | 90~95 | 95~99 |
> |:---------------------------------------:|:-----:|:-----:|:-----:|:-----:|:-----:|:-----:|:-----:|:-----:|:-----:|:-----:|:-----:|:-----:|
> |                    1                    | 0.871 | 0.945 | 0.967 | 0.982 | 0.990 | 0.995 | 0.996 | 0.998 | 0.999 | 0.999 | 1.000 | 1.000 |
> |                    2                    | 0.783 | 0.866 | 0.921 | 0.945 | 0.954 | 0.963 | 0.968 | 0.973 | 0.975 | 0.978 | 0.974 | 0.978 |
> |                    3                    | 0.701 | 0.803 | 0.852 | 0.893 | 0.906 | 0.923 | 0.923 | 0.929 | 0.937 | 0.929 | 0.922 | 0.926 |
> |                    4                    | 0.639 | 0.744 | 0.777 | 0.820 | 0.834 | 0.851 | 0.858 | 0.857 | 0.871 | 0.869 | 0.883 | 0.856 |
> |                    5                    | 0.592 | 0.657 | 0.707 | 0.750 | 0.766 | 0.779 | 0.789 | 0.793 | 0.799 | 0.799 | 0.790 | 0.787 |

---

> > ### Comment · Reviewer_fV1g · 2025-08-05
> >
> > I would like to thank the authors for their detailed responses. My concerns have been addressed, and I have therefore adjusted my rating to 4.

---

### Official Review · Reviewer_E8Tv · 2025-06-24

**Clarity:** 4
**Significance:** 4
**Originality:** 4
**Rating:** 6
**Confidence:** 5

**Summary:**

This submission presents a still image watermarking technique that is able to localize regions containing a watermark signal and decode the hidden message. There are two variants: 1. Embedding the watermark over the whole image. 2. Embedding the watermark in critical regions of the image. As a side product, the method is able to retrieve up to 5 different watermarks inside one image.

**Questions:**

- What explains the better performances (w.r.t. to all metrics - complexity, robustness, message lengths) compared to WAM?

- The editing processes included in the augmentation layer are stronger than the attacks considered for benchmarking robustness. Why? Although I agree that one does not need extreme robustness, I am wondering about the robustness limits of the proposed method. It would be a good idea to consider another setting with stronger attacks.

**Ethical Concerns:**

["NO or VERY MINOR ethics concerns only"]

**Final Justification:**

Thanks for the authors' clear reply. I have no more concerns. I am even more confident with keeping my score.

**Limitations:**

One limitation is that the watermark is visible. However, I am a watermark professional trained to spot these kinds of artefacts (without pretending to be a golden eye). This technique is typically not suitable when watermarking artworks (museum high definition scans, Hollywood movies etc).

The authors mention this limitation in App. A.
(BTW: 39dB is **not** a "*relatively high PSNR*", line 450)

Another limitation concerns reproducibility. The authors claim that "*information is sufficient to reproduce the main experimental results and verify the paper’s core claims, even without access to the code or data*". I have my doubts. The architecture is not detailed enough.
I acknowledge that the authors are willing to provide the code if accepted.

**Quality:**

4

**Strengths And Weaknesses:**

# Strengths

- S1. Very well written
- S2. Awesome performances
- S3. Beat WAM

# Weakness

- W1. Lack of detail about decoding multiple watermarks in one image. WAM has a specific procedure for this scenario, i.e. the DBSCAN clustering. The submission does not say anything.
- W2. A section detailing the main difference with WAM (architecture, training procedure...). At some point, it looks like too good to be true.

---

> ### Author Rebuttal · Authors · 2025-07-31
>
> # Author Response (Reviewer E8Tv)
> Dear Reviewer, thank you very much for your careful review of our paper and thoughtful comments. We hope the following responses can help address your concerns.
>
> ___
>
> **W1:** Lack of detail about decoding multiple watermarks in one image.
>
> **A1:** We appreciate your feedback. As described in Appendix C.2.4, MaskMark handles multiple watermarks by first predicting a segmentation mask that may contain multiple disjoint regions. To separate these, we apply OpenCV’s *cv2.connectedComponents*, which identifies and isolates each connected region. Each region is then decoded independently using the watermark decoder. This region-wise decoding approach parallels WAM's use of DBSCAN in purpose, although our segmentation-based method avoids the need for additional clustering.
>
> ___
>
> **W2&Q1:** A section detailing the main difference with WAM (architecture, training procedure...). What explains the better performances (w.r.t. to all metrics - complexity, robustness, message lengths) compared to WAM?
>
> **A:** Thank you for the suggestion. Below we summarize the main differences between MaskMark and WAM in terms of core design principles, model architecture, and training procedures, and explain why these differences lead to better performance across complexity, robustness, and message length.
>
> - **Core Design Principles.**
>     - The core idea of WAM is to **treat watermark extraction as a pixel-wise semantic segmentation problem**. Each pixel independently carries bits of the watermark, which are later averaged during inference. This creates several challenges:
>         - Extracting meaningful information from individual pixels is inherently difficult, leading to **slow convergence, high training complexity, and limited capacity for longer watermark messages** (as noted in the WAM paper).
>         - The final extraction relies on a **non-trainable and naive** voting/averaging mechanism across pixels, which cannot be optimized end-to-end and performs poorly when the watermarked region is small or per-pixel accuracy is low.
>     - In contrast, MaskMark is fundamentally built around spatially-aware encoding and decoding. The encoder and decoder are both guided by a spatial mask, enabling the system to:
>         - Embed and extract information across multiple pixels within a localized region, which **lowers training difficulty and allows extension to longer messages**.
>         - Train all components end-to-end, including both mask prediction and region-wise watermark embedding/extraction, avoiding WAM’s reliance on non-trainable inference heuristics. This leads to **improved robustness in local watermark extraction**
> - **Model Architecture.**
>     - The WAM architecture is heavily weighted toward the decoder, as its main challenge lies in extracting bits from individual pixels. It uses a VQGAN-based encoder and a ViT-Base decoder, resulting in a highly asymmetric and heavy design.
>     - MaskMark adopts the classical symmetric design commonly used in prior watermarking literature. Both the encoder and decoder use U-Net backbones, resulting in a lightweight and balanced architecture. Importantly:
>         - Unlike WAM, MaskMark decouples bit extraction and mask prediction. A lightweight $\text{U}^2$-Net is added in parallel to the decoder as a dedicated mask predictor, which improves optimization stability and convergence.
>         - This design is **backbone-agnostic** and highly flexible — MaskMark can easily adapt to different base architectures or be scaled up/down depending on the use case.
> - **Training Procedure.**
>     - WAM employs a **two-stage** training scheme: the first stage optimizes watermark extraction, while the second stage aims to improve the visual quality of the watermarked image. This is necessary because pixel-level bit decoding is extremely hard, and **optimizing both objectives simultaneously in the first stage can cause conflicts**. However, this adds substantial training overhead.
>     - MaskMark uses a **single-stage** training pipeline, similar to traditional watermarking methods. Both watermark extraction accuracy and visual quality are optimized jointly from the start. This leads to: **simplified training, faster convergence, and lower computational cost**.
>
> ___
>
> **Q2:** It would be a good idea to consider another setting with stronger attacks.
>
> **A2:** We appreciate your suggestion to consider a setting with stronger attacks.
> - In response, we increased the distortion strength beyond what the model was exposed to during training and conducted evaluation under these more challenging conditions.
> - As shown in Table 1 and Table 2, MaskMark consistently maintains strong extraction accuracy under these stronger attacks. This demonstrates the generalizability and robustness of our method, even when facing distortions that exceed the training distribution.
>
> **Table 1. Watermark extraction accuracy of MaskMark-D under stronger-than-training distortions across different watermarked area ratios.**
> |        Distortions / Watermarked Area (%)       |  1~5  |  5~10 | 10~20 | 20~30 | 30~40 | 40~50 | 50~60 | 60~70 | 70~80 | 80~90 | 90~95 | 95~99 |
> |:------------------------:|:-----:|:-----:|:-----:|:-----:|:-----:|:-----:|:-----:|:-----:|:-----:|:-----:|:-----:|:-----:|
> |        JPEG   (40)       | 0.631 | 0.779 | 0.870 | 0.919 | 0.955 | 0.970 | 0.979 | 0.985 | 0.989 | 0.991 | 0.993 | 0.995 |
> |  Gaussian Filter (2, 7)  | 0.895 | 0.988 | 0.999 | 1.000 | 1.000 | 1.000 | 1.000 | 1.000 | 1.000 | 1.000 | 1.000 | 1.000 |
> | Gaussian Noise (0, 0.13) | 0.787 | 0.941 | 0.990 | 0.998 | 0.999 | 1.000 | 1.000 | 1.000 | 1.000 | 1.000 | 1.000 | 1.000 |
> |     Median Filter (7)    | 0.871 | 0.982 | 0.998 | 0.999 | 1.000 | 1.000 | 1.000 | 1.000 | 1.000 | 1.000 | 1.000 | 1.000 |
> | Salt&Pepper Noise (0.13) | 0.549 | 0.639 | 0.785 | 0.893 | 0.950 | 0.976 | 0.990 | 0.995 | 0.997 | 0.998 | 0.999 | 0.999 |
> |  Rotation ([-100, 100])  | 0.543 | 0.603 | 0.680 | 0.754 | 0.801 | 0.837 | 0.868 | 0.904 | 0.929 | 0.944 | 0.942 | 0.949 |
> | Perspective ([0.1, 0.6]) | 0.590 | 0.667 | 0.742 | 0.807 | 0.849 | 0.881 | 0.903 | 0.937 | 0.948 | 0.959 | 0.965 | 0.975 |
>
> **Table 2. Watermark extraction accuracy of MaskMark-ED under stronger-than-training distortions across different watermarked area ratios.**
> |        Distortions / Watermarked Area (%)       |  1~5  |  5~10 | 10~20 | 20~30 | 30~40 | 40~50 | 50~60 | 60~70 | 70~80 | 80~90 | 90~95 | 95~99 |
> |:------------------------:|:-----:|:-----:|:-----:|:-----:|:-----:|:-----:|:-----:|:-----:|:-----:|:-----:|:-----:|:-----:|
> |        JPEG   (40)       | 0.842 | 0.962 | 0.990 | 0.997 | 0.998 | 0.999 | 0.999 | 0.999 | 0.999 | 0.999 | 0.999 | 0.999 |
> |  Gaussian Filter (2, 7)  | 0.943 | 0.996 | 0.999 | 1.000 | 1.000 | 1.000 | 1.000 | 1.000 | 1.000 | 1.000 | 1.000 | 1.000 |
> | Gaussian Noise (0, 0.13) | 0.905 | 0.988 | 0.999 | 1.000 | 1.000 | 1.000 | 1.000 | 1.000 | 1.000 | 1.000 | 1.000 | 1.000 |
> |     Median Filter (7)    | 0.925 | 0.994 | 0.999 | 1.000 | 1.000 | 1.000 | 1.000 | 1.000 | 1.000 | 1.000 | 1.000 | 1.000 |
> | Salt&Pepper Noise (0.13) | 0.680 | 0.864 | 0.954 | 0.985 | 0.993 | 0.996 | 0.998 | 0.998 | 0.998 | 0.999 | 0.999 | 0.999 |
> |  Rotation ([-100, 100])  | 0.564 | 0.643 | 0.720 | 0.792 | 0.837 | 0.882 | 0.909 | 0.928 | 0.943 | 0.947 | 0.965 | 0.965 |
> | Perspective ([0.1, 0.6]) | 0.643 | 0.741 | 0.807 | 0.875 | 0.913 | 0.931 | 0.946 | 0.960 | 0.968 | 0.967 | 0.973 | 0.975 |
>
> ___
>
> **L1:** One limitation is that the watermark is visible.
>
> **A1:** Thank you for pointing this out. As also acknowledged in Appendix A, we recognize that our current method may produce visible artifacts under close inspection by trained professionals. Addressing this limitation, especially for high-fidelity applications such as artworks or cinematic content, is part of our ongoing and future work.
>
> ___
>
> **L2:** Another limitation concerns reproducibility.
>
> **A2:** Thank you for raising this concern. We understand the importance of reproducibility and fully agree that providing code is essential for verifying our claims. If the paper is accepted, we will release **all code, pretrained models, and implementation details**, ensuring full reproducibility of our results.

---

### Official Review · Reviewer_oDY7 · 2025-06-30

**Clarity:** 3
**Significance:** 3
**Originality:** 3
**Rating:** 5
**Confidence:** 5

**Summary:**

This paper introduces MaskMark, a simple and efficient framework for image watermarking that supports both global and local watermark embedding, localization, and extraction through two variants: MaskMark-D and MaskMark-ED. It achieves state-of-the-art performance while being significantly faster than prior methods like WAM, and can be easily adapted to different robustness requirements.

**Questions:**

1. Following weakness 1.  Can the author provide ablation studies for investigating the functionality of different model components?
2. Similar to HiDDeN and other classical image watermarking methods, WAM learns a residual map and embeds it into the cover image at the pixel level, while MaskMark concatenates the watermark feature map with the cover image in the channel dimension. Is this concatenation a key design to ensure better performance for MaskMark?
3. Following weakness 2. Can the authors provide theoretical analysis or insights on why the proposed method can achieve better performance than the selected baselines?
4. WAM achieves multiple watermarks since the ViT decoder is a per-pixel-based message decoder, which can decode multiple watermarks simultaneously. Regarding MaskMark message decoder is one-dimensional bit-stream decoding. How can MaskMark achieve multiple watermarks under such settings?

**Ethical Concerns:**

["NO or VERY MINOR ethics concerns only"]

**Final Justification:**

The replies solve my concerns. Though there is no deeper theoretical analysis, the empirical results are very solid and are enough to explain the good performance. So I decided to maintain my rating.

**Limitations:**

yes

**Quality:**

3

**Strengths And Weaknesses:**

Strengths:
1. MaskMark achieves state-of-the-art performance in global and local watermark extraction, watermark localization, and multi-watermark embedding.
2. MaskMark provides two variants: MaskMark-D and MaskMark-ED for robustness. Both methods can support up to 128-bit message embedding.
3. The training of MaskMark is efficient to finish within one day, outperforming existing baselines such as WAM.
4. The paper is well-written and easy to follow, with a clear model architecture and training process, which helps facilitate reproducibility.

Weakness:
1. There is a lack of ablation studies, which limits the ability to identify the components for enhancing the model's performance.
2. There is a lack of theoretical analysis of why the proposed MaskMark method can outperform the existing localized watermarking baselines such as WAM and Editguard.

---

> ### Author Rebuttal · Authors · 2025-07-31
>
> # Author Response (Reviewer oDY7)
> Dear Reviewer, thank you very much for your careful review of our paper and thoughtful comments. We hope the following responses can help address your concerns.
>
> ___
>
> **W1&Q1:** Can the author provide ablation studies for investigating the functionality of different model components?
>
> **A:** Thank you for your great suggestion. We agree that ablation studies are important for understanding which components contribute to performance. In our case, MaskMark intentionally builds upon standard architectural components widely used in image watermarking. Specifically:
> - **Encoder: CNN + U-Net + JND.**
>     - The CNN + U-Net combination is a classical structure for watermark embedding [1][2][3][4] and we did not introduce any novel architectural modifications or attempt alternative variants here.
>     - The JND module, adopted from WAM, is used solely to enhance the visual quality of the watermarked image. We provide an ablation study on the effect of the JND module in Appendix D.1.
> - **Decoder: CNN + U-Net + $\text{U}^2$-Net.**
>     - Similarly, the CNN + U-Net stack is also a standard structure used for watermark extraction [1][4].
>     - The only additional component is a lightweight $\text{U}^2$-Net used for mask prediction. We explored different configurations, including varying the size of $\text{U}^2$-Net and replacing it with a larger U-Net, but observed minimal impact on performance. As such, we selected the small variant of $\text{U}^2$-Net for efficiency, which proved to be sufficiently effective.
>
> Overall, MaskMark’s strength lies not in the specific architectural choices, but in the mask mechanism itself. Because it is independent of the underlying backbone architecture, MaskMark remains flexible and easily transferable to other model designs.
>
> **References**
> 1. Sepmark: Deep separable watermarking for unified source tracing and deepfake detection. ACM MM, 2023.
> 2. Robust-wide: Robust watermarking against instruction-driven image editing. ECCV, 2024.
> 3. Trustmark: Universal watermarking for arbitrary resolution images. Arxiv, 2023.
> 4. Editguard: Versatile image watermarking for tamper localization and copyright protection. CVPR, 2024.
>
> ___
>
> **Q2:** Similar to HiDDeN and other classical image watermarking methods, WAM learns a residual map and embeds it into the cover image at the pixel level, while MaskMark concatenates the watermark feature map with the cover image in the channel dimension. Is this concatenation a key design to ensure better performance for MaskMark?
>
> **A2:** Thank you for this insightful observation. We would like to clarify that both the **residual map embedding** and the **channel-wise concatenation** you mentioned are standard practices in neural network-based image watermarking methods (including ours), and occur at different stages of the watermark embedding process. **Therefore, the channel-wise concatenation is not a key design choice unique to MaskMark, nor is it the main reason for its performance gains.**
> - **Residual map embedding** typically occurs after fusing the watermark information with the cover image to generate the encoded image, and **serves as the final step in producing the watermarked image**.
>     - Specifically, once the encoder outputs an encoded image, the residual map is computed as:
> $$I_\text{res} = I_\text{enc} − I_\text{cover},$$
> where $I_\text{res}$ is the residual map, $I_\text{enc}$ is the encoded image and $I_\text{cover}$ is the cover image.
> Then, the final watermarked image is produced by:
> $$I_\text{wm} = I_\text{cover} + \alpha \times I_\text{res}.$$
> where $I_\text{wm}$ is the watermarked image and $\alpha$ is a tunable strength factor balancing robustness and visual quality.
>     - This operation actually takes place inside the JND module in Figure 1 of the paper, rather than during the initial concatenation of $𝑓$. The residual map is further modulated by JND and then multiplied by the JND coefficient before being embedded into the cover image to produce the final watermarked image.
> - **Channel-wise concatenation**, on the other hand, **occurs at the early stage of the watermark encoder, before fusion begins**.
>     - Specifically, watermark information (either in the form of a repeated message map or an extracted feature map) is concatenated with the cover image or its intermediate features along the channel dimension.
>     - This fused tensor is then passed through the encoder (e.g., a U-Net) to produce the encoded image. This strategy is widely used in prior works such as HiDDeN, MBRS, and even WAM itself, and is **not a key design innovation of MaskMark**.
>
> ___
>
> **W2&Q3:** Can the authors provide theoretical analysis or insights on why the proposed method can achieve better performance than the selected baselines?
>
> **A:** Thank you for the insightful question. While our paper focuses on empirical results, we provide here a conceptual and architectural analysis of why MaskMark achieves superior performance compared to prior baselines across three categories: global watermarking, localized watermarking, and watermark localization methods.
>
> - **Compared to Global Watermarking Methods (e.g., StegaStamp, TrustMark).**
>     - Global watermarking methods are not designed to support localized watermark extraction, which is discussed in Lines 134–140 of the paper.
> - **Compared to WAM (Localized Watermarking Method).**
>     - The core idea of WAM is to **treat watermark extraction as a pixel-wise semantic segmentation problem**. Each pixel independently carries bits of the watermark, which are later averaged during inference. This creates several challenges:
>         - Extracting meaningful information from individual pixels is inherently difficult, leading to slow convergence, high training complexity, and limited capacity for longer watermark messages (as noted in the WAM paper).
>         - The final extraction relies on a **non-trainable and naive** voting/averaging mechanism across pixels, which cannot be optimized end-to-end and performs poorly when the watermarked region is small or per-pixel accuracy is low.
>     - In contrast, MaskMark is fundamentally built around spatially-aware encoding and decoding. The encoder and decoder are both guided by a spatial mask, enabling the system to:
>         - Embed and extract information across multiple pixels within a localized region, which **lowers training difficulty and allows extension to longer messages**.
>         - Train all components end-to-end, including both mask prediction and region-wise watermark embedding/extraction, avoiding WAM’s reliance on non-trainable inference heuristics. This leads to **improved robustness in local watermark extraction**
> - **Compared to EditGuard (Watermark Localization Method).**
>     - EditGuard embeds two separate types of signals: a 1D copyright watermark and a 2D fragile localization watermark to achieve watermark localization. This dual embedding comes with trade-offs:
>         - The addition of a second watermark signal increases visual distortion and complexity.
>         - Ensuring robustness of both the copyright and localization watermarks simultaneously is difficult and may involve conflicting optimization objectives, limiting the achievable performance.
>     - The watermark localized by MaskMark serves as a copyright watermark, which allows robustness and traceability to be ensured simultaneously. This leads to a higher potential performance upper bound and minimal impact on visual quality.
>
> ___
>
> **Q4:** WAM achieves multiple watermarks since the ViT decoder is a per-pixel-based message decoder, which can decode multiple watermarks simultaneously. Regarding MaskMark message decoder is one-dimensional bit-stream decoding. How can MaskMark achieve multiple watermarks under such settings?
>
> **A4:** Thank you for your question. While WAM uses a per-pixel ViT decoder to directly extract multiple watermarks, our approach handles multi-watermark decoding differently but effectively.
> - As detailed in Appendix C.2.4, MaskMark predicts a mask containing multiple spatially disjoint regions in a single forward pass. We then use OpenCV’s *cv2.connectedComponents* to segment this predicted mask into separate connected components, each corresponding to a distinct embedded watermark. Each segmented region is then processed independently by our one-dimensional bitstream decoder to extract its associated watermark message. This design enables MaskMark to support multiple watermarks simultaneously despite using a 1D decoder.

---

> > ### Comment · Reviewer_oDY7 · 2025-08-05
> >
> > Thanks for the authors' clear reply. I have no more concerns.

---

### Official Review · Reviewer_Umwi · 2025-06-30

**Clarity:** 3
**Significance:** 3
**Originality:** 2
**Rating:** 5
**Confidence:** 4

**Summary:**

The paper introduces MaskMark, a novel framework for image watermarking, designed to address limitations in existing methods. MaskMark offers two variants:

- MaskMark-D: This variant supports global watermark embedding, watermark localization (identifying where the watermark is present), and local watermark extraction. It achieves this by incorporating a simple masking mechanism during the decoding stage. During training, the decoder learns to localize and extract watermarks from specific regions based on various applied masks. MaskMark-D is particularly useful for applications like tamper detection.

- MaskMark-ED: This variant extends MaskMark-D by integrating the masking mechanism into both the encoding and decoding stages. This enables local watermark embedding and extraction, which significantly enhances robustness against regional attacks by guiding the encoder to embed the watermark in designated local regions.

MaskMark operates within the classical training paradigm of image watermarking. Through extensive experimentation, the authors demonstrate that MaskMark achieves state-of-the-art performance in global and local watermark extraction, watermark localization, and multi-watermark embedding. Notably, it outperforms leading existing baselines, including WAM (Watermark Anything Model). The framework also boasts high computational efficiency, being 15 times faster to train than WAM, and offers flexibility as its robustness can be fine-tuned by simply adjusting the distortion layer.

**Questions:**

- Clarify what part of the method was taken from WAM, what is novel. JND and the Lama masks come from WAM too right?
- What does "4. Lack of native local embedding: WAM embeds watermarks globally and then crops for local focus,
introducing embedding losses that reduce extraction robustness." mean?
- Is the method robust to moving a watermarked object inside the image or changing its proportions? WAM is.

**Ethical Concerns:**

["NO or VERY MINOR ethics concerns only"]

**Final Justification:**

see rebuttal

**Limitations:**

yes

**Paper Formatting Concerns:**

No.

**Quality:**

3

**Strengths And Weaknesses:**

Strengths

- State-of-the-Art Performance: MaskMark achieves superior performance in various watermarking tasks. It specifically outperforms strong baselines like WAM in local watermarking scenarios.

- High Computational Efficiency: training time significantly lower (15x faster) than that of WAM, and has fewer parameters

Weaknesses

- Limited Novelty: The idea is very similar to that of Watermark Anything in terms of goal, but also in terms of approach. For instance, the authors use the masks from Lama similarly to what is done in WAM. Overall it appears like a WAM+, but with limited novelty, or at least not highlighted enough.

---

> ### Author Rebuttal · Authors · 2025-07-31
>
> # Author Response (Reviewer Umwi)
> Dear Reviewer, thank you very much for your careful review of our paper and thoughtful comments. We hope the following responses can help clarify potential misunderstandings and alleviate your concerns.
>
> ___
>
> **W1&Q1:** The idea is very similar to that of Watermark Anything in terms of goal, but also in terms of approach. Clarify what part of the method was taken from WAM, what is novel.
>
> **A:** We appreciate your concerns and the opportunity to clarify the novelty and positioning of our method relative to WAM.
>
> It is true that the only components we share with WAM are the JND module and the use of LaMa masks.
> - These choices were indeed inspired by WAM; however, they are neither core components nor novel contributions of either method.
> - Both are long-standing tools commonly used in image editing [1][2][3] and watermarking research [4][5][6], and we incorporated them purely as standard building blocks.
>
> Our key contributions go well beyond these shared elements, in both functionality and architectural design:
>
> - **Native Local Watermark Embedding: A Functional Innovation.**
>     - Unlike WAM, which lacks native support for local watermark embedding, MaskMark is specifically designed for this functionality. While WAM embeds a global watermark and then applies post-hoc cropping to simulate local watermark embedding (which introduces signal loss and robustness issues), MaskMark embeds the watermark directly into designated regions by conditioning encoder on a spatial mask. This results in **improved robustness and precise regional protection**.
> - **Fundamental Differences in Core Design Philosophy.**
>     - The core idea of WAM is to **treat watermark extraction as a pixel-wise semantic segmentation problem**. Each pixel independently carries bits of the watermark, which are later averaged during inference. This creates several challenges:
>         - Extracting meaningful information from individual pixels is inherently difficult, leading to **slow convergence, high training complexity, and limited capacity for longer watermark messages** (as noted in the WAM paper).
>         - The final extraction relies on a **non-trainable and naive** voting/averaging mechanism across pixels, which cannot be optimized end-to-end and performs poorly when the watermarked region is small or per-pixel accuracy is low.
>     - In contrast, MaskMark is fundamentally built around **spatially-aware encoding and decoding**. The encoder and decoder are both guided by a spatial mask, enabling the system to:
>         - Embed and extract information across multiple pixels within a localized region, which **lowers training difficulty and allows extension to longer messages**.
>         - Train all components end-to-end, including both mask prediction and region-wise watermark embedding/extraction, avoiding WAM’s reliance on non-trainable inference heuristics. This leads to **improved robustness in local watermark extraction**.
>
> In summary, while we acknowledge WAM as an important prior work, MaskMark introduces a functionally novel capability (native local embedding) and is built upon a substantially different architectural philosophy. These distinctions are crucial for achieving improved efficiency and robustness.
>
> **References**
> 1. Instructdiffusion: A generalist modeling interface for vision tasks. CVPR, 2024.
> 2. Paint by example: Exemplar-based image editing with diffusion models. CVPR, 2023.
> 3. Inpaint anything: Segment anything meets image inpainting. Arxiv, 2023.
> 4. Color image watermarking based on orientation diversity and color complexity. Expert Systems with Applications, 2020.
> 5. Pattern complexity-based JND estimation for quantization watermarking. Pattern Recognition Letters, 2020.
> 6. Towards perceptual image watermarking with robust texture measurement. Expert Systems with Applications, 2023.
>
> ___
>
> **Q2:** What does "4. Lack of native local embedding: WAM embeds watermarks globally and then crops for local focus, introducing embedding losses that reduce extraction robustness." mean?
>
> **A2:** Thank you for the question. This comment refers to a limitation of WAM in supporting localized watermark embedding.
> - WAM performs watermark embedding over the entire image globally. To simulate a local watermark, WAM must embed a global watermark first and then crop the image to retain only the protected region. The remaining (unprotected) regions are replaced post hoc with the corresponding clean (non-watermarked) content from the original image.
> - This process inherently causes the loss of watermark signal in cropped-out regions and creates inconsistencies at the region boundaries, which ultimately reduces the robustness and reliability of watermark extraction.
>
> ___
>
> **Q3:** Is the method robust to moving a watermarked object inside the image or changing its proportions?
>
> **A3:** Thank you for raising this interesting and challenging attack scenario that combines both valuemetric and geometric distortions, namely, the MoveAndResize attack.
> - We have evaluated this setting as follows: a watermarked region (as defined by the segment mask) is cropped and resized using a random scale factor sampled from the range (0.5, 1.2). This resized patch is then pasted onto a different, non-watermarked image, and watermark extraction is performed on the composite.
> - Evaluation results for MaskMark-D, MaskMark-ED, and WAM under this setting are shown in Table 1 below.
>     - We observe that when the watermarked region is small, the default version of MaskMark may underperform. However, as the region size increases, MaskMark consistently outperforms WAM.
>     - By incorporating the MoveAndResize distortion into the training pipeline and fine-tuning the model for just 50k steps, we achieved significant improvements in robustness (see the FT version in Table 1). In this adapted setting, MaskMark surpasses WAM across all region sizes.
>     - WAM appears to suffer from a performance bottleneck when dealing with large watermarked regions. This is due to limited performance of its mask prediction, which hinders complete and precise recovery of the watermarked region for extraction. In contrast, MaskMark does not exhibit this issue.
>
> **Table 1. Watermark extraction accuracy under the MoveAndResize attack across different watermarked area ratios.**
> |      Model / Watermarked Area (%)     |  1~5  |  5~10 | 10~20 | 20~30 | 30~40 | 40~50 | 50~60 | 60~70 | 70~80 | 80~90 | 90~95 | 95~99 |
> |:--------------:|:-----:|:-----:|:-----:|:-----:|:-----:|:-----:|:-----:|:-----:|:-----:|:-----:|:-----:|:-----:|
> |       WAM      | 0.541 | 0.601 | 0.677 | 0.722 | 0.743 | 0.770 | 0.762 | 0.744 | 0.791 | 0.751 | 0.758 | 0.760 |
> |   MaskMark-D   | 0.519 | 0.533 | 0.584 | 0.652 | 0.675 | 0.723 | 0.775 | 0.769 | 0.834 | 0.824 | 0.852 | 0.854 |
> |  MaskMark-D-FT | 0.551 | 0.627 | 0.742 | 0.848 | 0.895 | 0.945 | 0.973 | 0.980 | 0.989 | 0.996 | 0.996 | 0.998 |
> |   MaskMark-ED  | 0.523 | 0.549 | 0.613 | 0.677 | 0.700 | 0.769 | 0.803 | 0.809 | 0.860 | 0.849 | 0.857 | 0.870 |
> | MaskMark-ED-FT | 0.662 | 0.787 | 0.896 | 0.960 | 0.977 | 0.992 | 0.996 | 0.998 | 0.999 | 0.999 | 0.999 | 1.000 |

---

> > ### Comment · Reviewer_Umwi · 2025-08-07
> >
> > Dear authors,
> >
> >  Thank you for the rebuttal which addresses most of my concerns. I will raise my score.

---

### Official Review · Reviewer_4ngb · 2025-07-02

**Clarity:** 3
**Significance:** 3
**Originality:** 2
**Rating:** 5
**Confidence:** 4

**Summary:**

The authors propose an image watermarking method that allows for localized message embedding and detection. In contrast to the prior work Watermark Anything, which performs binary pixel-level detection and message decoding followed by clustering on decoded messages to localize watermarked regions (meaning localization emerges as a consequence of training for pixel-level detection/decoding), the proposed method is trained to natively perform localization through the use of masks.

Training:

* After embedding, random masks are applied to limit the watermark to specific image regions while preserving the original image content outside those regions.
* At detection time, the detector is passed both a masked image containing only the watermarked region (from which it is trained to decode the message) and the entire partially-watermarked image (from which it is trained to predict a localization mask).
* In the MaskMark-D variant, the embedder is not aware of the mask and thus must learn to embed global information that is still recoverable from (potentially small) masked regions. In the MaskMark-ED variant, the embedder is aware of the mask and thus learns to concentrate watermark information more locally and embed additional information to aid localization.


Inference:

* The detector first predicts a localization mask; this mask is then applied to the image, which is decoded to obtain the watermark message

The proposed method outperforms existing post-hoc image watermarking methods, including Watermark Anything, in watermark localization and in accurately extracting local watermark messages under standard image distoritons.

**Questions:**

* Can the authors more clearly indicate which evaluated image distortions were not seen by the model at training time? This kind of generalizability is important for evaluating the robustness of watermarking methods

**Ethical Concerns:**

["NO or VERY MINOR ethics concerns only"]

**Final Justification:**

In their rebuttal, the authors addressed many of the concerns raised in my review. The additional experiments against purification attacks show higher than expected robustness. While it would be nice to see additional experiments with actual "adaptive" attacks that leverage knowledge of the specific method, clarifying the terminology is a good step. Overall, I feel that this work will be of interest to researchers working on post-hoc localized watermarking.

**Limitations:**

Yes.

**Paper Formatting Concerns:**

None.

**Quality:**

3

**Strengths And Weaknesses:**

__Strengths__

* The proposed method is an interesting alternative to the localization scheme from Watermark Anything.

* The proposed method performs competitively with or better than many existing post-hoc methods in global embedding (Table 1) and local embedding (Figure 3), especially with small watermarked regions. This does not seem to come with a significant tradeoff in SSIM/PSNR, on which the proposed method outperforms Watermark Anything.

* The proposed method does not require an adversarial discriminator or "perceptual" loss for the embedder, just a simple MSE loss. The proposed JND modulation approach appears to effectively mitigate the perceptible artifacts that arise from MSE-only training, as illustrated in Appendix D.

* Overall, the proposed approach seems to present a straightforward improvement over Watermark Anything, which as far as I can tell is considered state of the art for localized image watermarking.


__Weaknesses__

* Terminology: I'm not sure that VAE attacks can be considered "adaptive". This term was coined for adversarial attacks performed with knowledge of an adversarial defense (https://arxiv.org/abs/2002.08347, NeurIPS 2020). However, the authors here appear to be fine-tuning their watermark against specific VAE attacks and then evaluating on these same attacks, which is the opposite of "adaptive" -- the defense is being conducted with specific knowledge of the attack, not the other way around! "Adaptive" attacks would leverage specific knowledge of the watermarking method, and could include e.g. adversarial optimization-based attacks against a surrogate or actual detector network.

* The proposed method presumably shares the inherent robustness limitations of existing post-hoc image watermarks, which are vulnerable against generative "purification" attacks (https://arxiv.org/abs/2306.01953, NeurIPS 2024). These differ somewhat from the VAE attacks considered by the authors in that they typically perform partial noising/denoising with a diffusion model, and have been shown to be effective in removing both watermarks and other "imperceptible" image perturbations (https://arxiv.org/abs/2406.12027). It is discouraging that the authors apparently considered a small number of VAE attacks (with the advantage of fine-tuning) without considering known stronger attacks.

* There is currently another published watermarking method with the name MaskMark [1]. On the one hand, the other work is in the audio domain rather than the image domain. On the other hand, both methods are neural network-based post-hoc watermarks that leverage some form of "masking," end-to-end training, distortion layer, etc. I would strongly recommend a name change.


[1] Patrick O’Reilly, Zeyu Jin, Jiaqi Su, and Bryan Pardo. Maskmark: Robust neural watermarking for real and synthetic speech. In International Conference on Acoustics, Speech and Signal Processing (ICASSP), 2024

---

> ### Author Rebuttal · Authors · 2025-07-31
>
> # Author Response (Reviewer 4ngb)
> Dear reviewer, thank you very much for your careful review of our paper and thoughtful comments. We hope the following responses can help address your concerns.
>
> ___
>
> **W1:** I'm not sure that VAE attacks can be considered "adaptive".
>
> **A1:** Thank you for pointing this out! We agree that our use of the term "adaptive" was inappropriate in this context, and we appreciate the opportunity to clarify.
> - What we intended to convey is that users can adapt the model to specific attack scenarios by fine-tuning it with particular distortions included in the distortion layer. In this sense, the model becomes robust to specific attacks through targeted fine-tuning.
> - Therefore, the term "adaptive" here refers to the model being fine-tuned to adaptively enhance its robustness against specific attacks, rather than describing the attacks themselves as adaptive. This is a writing issue on our part, we should not have placed "adaptive" in front of "attacks."
> - To avoid confusion, we will revise our terminology and replace "adaptive" with "specific" when referring to this type of robustness enhancement via fine-tuning.
>
> ___
>
> **W2:** The proposed method presumably shares the inherent robustness limitations of existing post-hoc image watermarks, which are vulnerable against generative "purification" attacks.
>
> **A2:** Thank you for highlighting this important class of attacks. We agree that generative “purification” attacks, such as partial noising/denoising using diffusion models, pose a serious challenge for post-hoc watermarking methods.
> - To address this, we conducted evaluations of both the default and VAE fine-tuned versions of MaskMark-D and MaskMark-ED under such purification attacks. Specifically, we used Stable Diffusion v1.5 with an empty prompt, applied 20 steps of forward noising to the watermarked image, and then performed denoising to generate the purified output.
> - As shown in Table 1, the default model (without fine-tuning) is indeed vulnerable to such generative purification. However, the VAE fine-tuned variant demonstrates strong robustness even against this unseen purification attack, indicating great generalization.
> - We hypothesize that this is due to the underlying similarity between VAE-based regeneration attacks and diffusion-based purification: they both aim to reconstruct the input while removing imperceptible signals. As such, incorporating differentiable VAE corruption during training provides a practical and effective proxy that enhances robustness against a broader class of generative purification attacks.
>
> **Table 1. Watermark extraction accuracy under generative "purification" attacks across different watermarked area ratios.**
> |        Model / Watermarked Area (%)       |  1~5  |  5~10 | 10~20 | 20~30 | 30~40 | 40~50 | 50~60 | 60~70 | 70~80 | 80~90 | 90~95 | 95~99 |
> |:------------------:|:-----:|:-----:|:-----:|:-----:|:-----:|:-----:|:-----:|:-----:|:-----:|:-----:|:-----:|:-----:|
> |     MaskMark-D     | 0.507 | 0.528 | 0.566 | 0.602 | 0.627 | 0.637 | 0.667 | 0.671 | 0.685 | 0.689 | 0.695 | 0.715 |
> |  MaskMark-D-VAE-FT | 0.515 | 0.582 |  0.67 | 0.753 | 0.811 | 0.843 | 0.872 | 0.898 | 0.924 | 0.934 | 0.945 | 0.947 |
> |     MaskMark-ED    | 0.542 | 0.606 | 0.639 | 0.658 | 0.652 | 0.661 | 0.653 | 0.644 | 0.625 | 0.610 | 0.599 | 0.581 |
> | MaskMark-ED-VAE-FT | 0.616 | 0.750 | 0.838 | 0.905 | 0.928 | 0.945 | 0.947 | 0.960 | 0.969 | 0.967 | 0.974 | 0.976 |
>
> ___
>
> **W3:** I would strongly recommend a name change.
>
> **A3:** Thank you for your great suggestion! We were not aware of the name overlap with the audio watermarking method. We appreciate your suggestion and will revise the name in the next revision or the camera-ready version to avoid confusion.
>
> ___
>
> **Q1:** Can the authors more clearly indicate which evaluated image distortions were not seen by the model at training time?
>
> **A1:** Thank you for highlighting this important aspect of robustness evaluation. We agree that assessing generalizability to unseen distortions is critical for a thorough understanding of a watermarking method’s reliability.
> - We list in Table 2 and Table 3 the distortion types that were not seen during training, along with their corresponding results, as originally reported in the main paper.
> - To further address your concern, we conducted additional evaluations (Table 4 and Table 5), where we increased the intensity of certain test-time distortions beyond the levels seen during training. Although these distortions are of the same types, their extreme variants were never encountered by the model, and thus can be considered unseen in a broader sense.
> - Across both types of evaluation, MaskMark demonstrates strong robustness, indicating its generalizability to distortions beyond those encountered during training.
>
> **Table 2. Watermark extraction accuracy of MaskMark-D under distortions not seen during the training stage across different watermarked area ratios.**
> |       Distortions / Watermarked Area (%)       |  1~5  |  5~10 | 10~20 | 20~30 | 30~40 | 40~50 | 50~60 | 60~70 | 70~80 | 80~90 | 90~95 | 95~99 |
> |:-----------------------:|:-----:|:-----:|:-----:|:-----:|:-----:|:-----:|:-----:|:-----:|:-----:|:-----:|:-----:|:-----:|
> |      Resize   (0.5)     | 0.907 | 0.991 | 0.999 | 1.000 | 1.000 | 1.000 | 1.000 | 1.000 | 1.000 | 1.000 | 1.000 | 1.000 |
> | Brightness ([0.7, 1.3]) | 0.896 | 0.989 | 0.999 | 1.000 | 1.000 | 1.000 | 1.000 | 1.000 | 1.000 | 1.000 | 1.000 | 1.000 |
> |  Contrast ([0.7, 1.3])  | 0.905 | 0.991 | 0.999 | 1.000 | 1.000 | 1.000 | 1.000 | 1.000 | 1.000 | 1.000 | 1.000 | 1.000 |
> |    Hue ([-0.1, 0.1])    | 0.821 | 0.950 | 0.987 | 0.999 | 0.998 | 0.999 | 0.999 | 0.999 | 0.999 | 0.999 | 1.000 | 1.000 |
> | Saturation ([0.7, 1.3]) | 0.897 | 0.988 | 0.999 | 1.000 | 1.000 | 1.000 | 1.000 | 1.000 | 1.000 | 1.000 | 1.000 | 1.000 |
>
> **Table 3. Watermark extraction accuracy of MaskMark-ED under distortions not seen during the training stage across different watermarked area ratios.**
> |       Distortions / Watermarked Area (%)       |  1~5  |  5~10 | 10~20 | 20~30 | 30~40 | 40~50 | 50~60 | 60~70 | 70~80 | 80~90 | 90~95 | 95~99 |
> |:-----------------------:|:-----:|:-----:|:-----:|:-----:|:-----:|:-----:|:-----:|:-----:|:-----:|:-----:|:-----:|:-----:|
> |      Resize   (0.5)     | 0.947 | 0.997 | 0.999 | 1.000 | 1.000 | 1.000 | 1.000 | 1.000 | 1.000 | 1.000 | 1.000 | 1.000 |
> | Brightness ([0.7, 1.3]) | 0.943 | 0.997 | 0.999 | 1.000 | 1.000 | 1.000 | 1.000 | 1.000 | 1.000 | 1.000 | 1.000 | 1.000 |
> |  Contrast ([0.7, 1.3])  | 0.947 | 0.997 | 1.000 | 1.000 | 1.000 | 1.000 | 1.000 | 1.000 | 1.000 | 1.000 | 1.000 | 1.000 |
> |    Hue ([-0.1, 0.1])    | 0.894 | 0.983 | 0.996 | 0.999 | 0.999 | 0.999 | 0.999 | 0.999 | 0.999 | 0.997 | 0.999 | 0.997 |
> | Saturation ([0.7, 1.3]) | 0.942 | 0.996 | 0.999 | 1.000 | 1.000 | 1.000 | 1.000 | 1.000 | 1.000 | 1.000 | 1.000 | 1.000 |
>
> **Table 4. Watermark extraction accuracy of MaskMark-D under stronger-than-training distortions across different watermarked area ratios.**
> |        Distortions / Watermarked Area (%)       |  1~5  |  5~10 | 10~20 | 20~30 | 30~40 | 40~50 | 50~60 | 60~70 | 70~80 | 80~90 | 90~95 | 95~99 |
> |:------------------------:|:-----:|:-----:|:-----:|:-----:|:-----:|:-----:|:-----:|:-----:|:-----:|:-----:|:-----:|:-----:|
> |        JPEG   (40)       | 0.631 | 0.779 | 0.870 | 0.919 | 0.955 | 0.970 | 0.979 | 0.985 | 0.989 | 0.991 | 0.993 | 0.995 |
> |  Gaussian Filter (2, 7)  | 0.895 | 0.988 | 0.999 | 1.000 | 1.000 | 1.000 | 1.000 | 1.000 | 1.000 | 1.000 | 1.000 | 1.000 |
> | Gaussian Noise (0, 0.13) | 0.787 | 0.941 | 0.990 | 0.998 | 0.999 | 1.000 | 1.000 | 1.000 | 1.000 | 1.000 | 1.000 | 1.000 |
> |     Median Filter (7)    | 0.871 | 0.982 | 0.998 | 0.999 | 1.000 | 1.000 | 1.000 | 1.000 | 1.000 | 1.000 | 1.000 | 1.000 |
> | Salt&Pepper Noise (0.13) | 0.549 | 0.639 | 0.785 | 0.893 | 0.950 | 0.976 | 0.990 | 0.995 | 0.997 | 0.998 | 0.999 | 0.999 |
> |  Rotation ([-100, 100])  | 0.543 | 0.603 | 0.680 | 0.754 | 0.801 | 0.837 | 0.868 | 0.904 | 0.929 | 0.944 | 0.942 | 0.949 |
> | Perspective ([0.1, 0.6]) | 0.590 | 0.667 | 0.742 | 0.807 | 0.849 | 0.881 | 0.903 | 0.937 | 0.948 | 0.959 | 0.965 | 0.975 |
>
> **Table 5. Watermark extraction accuracy of MaskMark-ED under stronger-than-training distortions across different watermarked area ratios.**
> |        Distortions / Watermarked Area (%)       |  1~5  |  5~10 | 10~20 | 20~30 | 30~40 | 40~50 | 50~60 | 60~70 | 70~80 | 80~90 | 90~95 | 95~99 |
> |:------------------------:|:-----:|:-----:|:-----:|:-----:|:-----:|:-----:|:-----:|:-----:|:-----:|:-----:|:-----:|:-----:|
> |        JPEG   (40)       | 0.842 | 0.962 | 0.990 | 0.997 | 0.998 | 0.999 | 0.999 | 0.999 | 0.999 | 0.999 | 0.999 | 0.999 |
> |  Gaussian Filter (2, 7)  | 0.943 | 0.996 | 0.999 | 1.000 | 1.000 | 1.000 | 1.000 | 1.000 | 1.000 | 1.000 | 1.000 | 1.000 |
> | Gaussian Noise (0, 0.13) | 0.905 | 0.988 | 0.999 | 1.000 | 1.000 | 1.000 | 1.000 | 1.000 | 1.000 | 1.000 | 1.000 | 1.000 |
> |     Median Filter (7)    | 0.925 | 0.994 | 0.999 | 1.000 | 1.000 | 1.000 | 1.000 | 1.000 | 1.000 | 1.000 | 1.000 | 1.000 |
> | Salt&Pepper Noise (0.13) | 0.680 | 0.864 | 0.954 | 0.985 | 0.993 | 0.996 | 0.998 | 0.998 | 0.998 | 0.999 | 0.999 | 0.999 |
> |  Rotation ([-100, 100])  | 0.564 | 0.643 | 0.720 | 0.792 | 0.837 | 0.882 | 0.909 | 0.928 | 0.943 | 0.947 | 0.965 | 0.965 |
> | Perspective ([0.1, 0.6]) | 0.643 | 0.741 | 0.807 | 0.875 | 0.913 | 0.931 | 0.946 | 0.960 | 0.968 | 0.967 | 0.973 | 0.975 |

---

> > ### Comment · Reviewer_4ngb · 2025-08-05
> > **Reply to Authors**
> >
> > I thank the authors for their detailed reply and additional experiments addressing the concerns raised in my review. In light of this I have raised my score. Overall, I think the proposed method shows meaningful improvement over previous localized post-hoc watermarking methods while remaining conceptually simple and straightforward to implement.

---

### Decision · Program_Chairs · 2025-09-17

**Decision:**

Accept (poster)

**Comment:**

# Summary

This paper presents MaskMark, an image watermarking framework that enables localized message embedding and detection. In contrast to previous approaches, MaskMark is explicitly trained to perform native localization through the use of spatial masks.

The authors propose two variants of the method, differing in whether the embedder is informed of the mask. When informed, the embedder can focus the watermark on specific regions, thereby improving robustness against localized attacks.

At inference time, the detector predicts a localization mask, applies it to the image, and then extracts the watermark from the masked region.

Extensive experiments demonstrate that MaskMark outperforms existing methods in terms of localization accuracy, both local and global message extraction, robustness to image distortions, and support for multiple watermarks.

# Recommendation

All reviewers were enthusiastic about the paper, and I share their positive assessment. I therefore recommend acceptance.

I encourage the authors to incorporate the reviewers’ feedback and points raised during the rebuttal discussion when preparing the final version of the paper.